# Chemoselective cycloisomerization of O-alkenylbenzamides via concomitant 1,2-aryl migration/elimination mediated by hypervalent iodine reagents

Jiaxin He[1,3], Feng-Huan Du [2,3], Chi Zhang [2✉] & Yunfei Du [1✉]

As an ambient nucleophile, controlling the reaction selectivities of nitrogen and oxygen atoms in amide moiety is a challenging issue in organic synthesis. Herein, we present a chemodivergent cycloisomerization approach to construct isoquinolinone and iminoisocoumarin skeletons from o-alkenylbenzamide derivatives. The chemo-controllable strategy employed an exclusive 1,2-aryl migration/elimination cascade, enabled by different hypervalent iodine species generated in situ from the reaction of iodosobenzene (PhIO) with MeOH or 2,4,6-tris-isopropylbenzene sulfonic acid. DFT studies revealed that the nitrogen and oxygen atoms of the intermediates in the two reaction systems have different nucleophilicities and thus produce the selectivity of N or O-attack modes.

[1] Tianjin Key Laboratory for Modern Drug Delivery & High-Efficiency, School of Pharmaceutical Science and Technology, Tianjin University, 300072 Tianjin, China. [2] State Key Laboratory of Elemento-Organic Chemistry, The Research Institute of Elemento-Organic Chemistry, College of Chemistry, Nankai University, 300071 Tianjin, China. [3] These authors contributed equally: Jiaxin He, Feng-Huan Du. ✉email: zhangchi@nankai.edu.cn; duyunfeier@tju.edu.cn

Organic compounds bearing an amide group are significant building blocks for the synthesis of pharmaceuticals and natural products[1–4]. As an ambident nucleophile, the peculiar reactivities of nitrogen and oxygen atoms in amide moiety diversify the reaction processes and simultaneously enhance the difficulty of controlling the reaction selectivities. Generally, the reaction chemoselectivities of ambident amide compounds can be controlled by the utilization of appropriate catalysts and reagents[5–9]. For examples, Nishikata's group has reported an intermolecular cyclization of α-bromoamides and acrylates, with the reactivity of the nitrogen or oxygen nucleophile of the amide group being finely controlled by using a copper catalyst with an appropriate base (Fig. 1a)[10]. Adopting o-alkynylbenzamides as substrate, Belmont and coworkers demonstrated the divergent synthesis of five-membered heterocyclic isoindolinones and isobenzofuranones by silver-catalyzed cycloisomerization[11]. However, differing from Belmont's work, Jiang and colleagues realized the divergent synthesis of six-membered isoquinolinone and iminoisocoumarin derivatives, the reaction of which was controlled by using gold/ligand or platinum catalyst (Fig. 1b)[12]. Despite these excellent advances, to the best of our knowledge, no document describing the control of the ambident reactivities of o-alkenylbenzamides has been reported till now. In this regard, a new protocol dictating the chemodivergent cycloisomerization of o-alkenylbenzamides, by tuning the differentiation of the N vs O nucleophilic strength, to construct novel heterocyclic skeletons should be highly desirable.

Over the past decades, hypervalent iodine reagents (HIRs) have attracted great attention in the field of organic synthesis, owing to their mild, environmentally benign characteristics and chemoselective oxidizing properties in contrast to heavy metal-based oxidants[13–20]. Inspired by the exclusive rearrangement[21–31] reaction mediated by hypervalent iodine reagents and our previous work about constructing isocoumarins derivatives via 1,2-aryl migration tactics[32], we envisioned that the reaction of hypervalent iodine reagents and o-alkenylbenzamides might trigger 1,2-aryl migration processes[33–42] and thus bring about an opportunity to differentiate nucleophilic sites. Herein, we present our results in controlling the reactivity of o-alkenylbenzamides by hypervalent iodine species generated in situ, which realized the divergent synthesis of isoquinolinones[43–46] and iminoisocoumarins[47–50] derivatives (Fig. 1c). It should be noted that there are no precedents on hypervalent iodine controlled chemoselectivity in the cyclization of o-alkenylbenzamides, and the work described here highlights a one-pot transformation involving an exclusive cascade sequence of chemoselective cyclization, 1,2-aryl migration and elimination processes.

## Results and discussion

Our initial studies focused on the optimization of reaction conditions. At the outset, o-alkenylbenzamide 1a was selected as the model substrate to examine the cycloisomerization reaction, and the results were listed in Table 1. When substrate 1a was treated with iodosobenzene (PhIO) in MeOH, combined with BF$_3$·OEt$_2$ (0.2 equiv) as a Lewis acid, we found the reaction displayed a completely distinct N-attack cyclization mode to furnish isoquinolinone 2a in 56% yield. After screening the effect acidic or basic of activators (entries 2–7), we found that TMSOTf was the most efficient catalyst for this transformation (Table 1, entry 5). Further investigation on reaction temperature revealed that the best outcome was obtained when the reaction was operated at reflux temperature (Table 1, entry 8). On the basis of the above screening results, the most optimal conditions for converting 1a to isoquinolinone 2a were concluded to be: substrate 1a with PhIO (1.5 equiv) and TMSOTf (0.2 equiv) in MeOH at reflux temperature.

Encouraged by the above results, we then turned our attention toward exploring the chemodivergent pattern of the protocol for the synthesis of O-cyclized iminoisocoumarin products. To our delight, we discovered that when substrate 1a combined with hydroxy(tosyloxy)iodobenzene (HTIB) in DCE (1,2-dichloroethane) at ambient temperature for 0.5 h, the reaction resulted in the formation of iminoisocoumarin 3a in 44% yield (Table 1, entry 9). With the encouraging result in hand, we came to further optimize the reaction conditions. First, a detailed screening of solvents, hypervalent iodine reagents and temperature was carried out (Supplementary Table S2). The result indicated that the reaction of substrate 1a with HTIB at 80 °C in DCE (1,2-dichloroethane) provided iminoisocoumarin 3a with 55% yield, with the reaction completed within 5 min (Table 1, entry 10). Next, almost identical results were obtained when using active hypervalent iodine species formed in situ from iodosobenzene (PhIO) and 4-toluenesulfonic acid (Table 1, entry 11)[51,52]. Furthermore, sulfonic acids with different structures were further tested (Table 1, entries 12–14), and 2,4,6-tris-isopropylbenzene sulfonic acid (S4) was identified as the most efficient promoter. To our delight, better yields were further achieved with the addition of an exogenous Lewis acid (Table 1, entries 15–18). Specifically, lithium perchlorate (LiClO$_4$) was found to be the optimal additive as the reaction gave the desired product 3a in 90% isolated yield. Further investigation on the loading of LiClO$_4$ indicated that neither decreasing nor increasing the equivalents were beneficial (Table 1, entries 19–20). Based on the above results, the optimal conditions for the transformation to iminoisocoumarin 3a were finalized to be: PhIO (1.5 equiv), 2,4,6-tris-isopropylbenzene sulfonic acid (S4; 1.5 equiv), LiClO$_4$ (1.5 equiv) in DCE at 80 °C.

With the optimized conditions in hand, we began to explore the general applicability of the divergent transformation, targeting the N-cyclization by using variously substituted N-phenyl-2-alkenylbenzamides with MeOH as a reaction partner[34,53–57] and solvent being first studied (Fig. 2). Replacing the methyl group

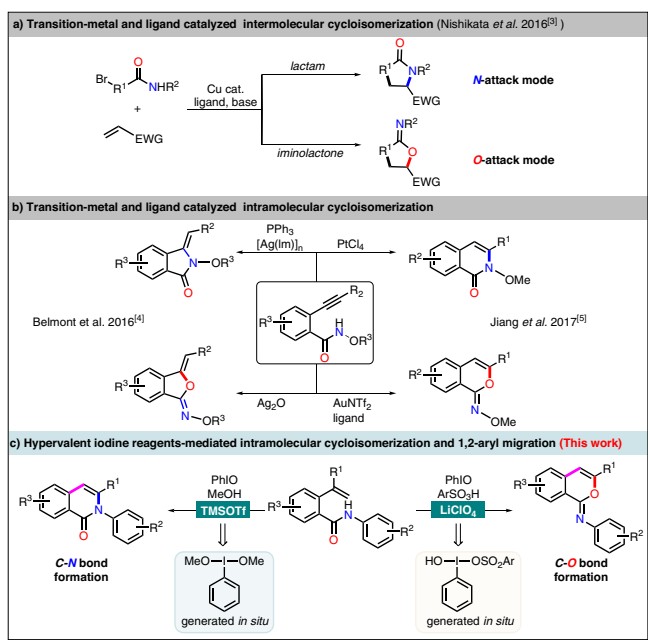

**Fig. 1 Strategies for divergent cyclization of amide derivatives.**
a, b Previous works on cycloisomerization of amide derivatives. c This work, hypervalent iodine reagents mediated intramolecular cycloisomerization and 1,2-aryl migration.

**Table 1 Optimization of the reaction conditions[a].**

| Entry | HIR | Solvent | Additive[b] | T (°C) | Yield[c] of 2a (%) | 3a (%) |
|---|---|---|---|---|---|---|
| 1 | PhIO | MeOH | BF₃•OEt₂ | rt | 56 | — |
| 2 | PhIO | MeOH | Et₃N | rt | NR | — |
| 3 | PhIO | MeOH | TFA | rt | 61 | — |
| 4 | PhIO | MeOH | TfOH | rt | 63 | — |
| 5 | PhIO | MeOH | TMSOTf | rt | 70 | — |
| 6 | PhIO | MeOH | LiClO₄ | rt | NR | — |
| 7 | PhIO | MeOH | 50%H₂SO₄ | rt | 58 | — |
| 8 | PhIO | MeOH | TMSOTf | Reflux | 81 | — |
| 9 | HTIB | DCE | — | rt | — | 44 |
| 10 | HTIB | DCE | — | 80 | — | 55 |
| 11 | PhIO | DCE | S1 | 80 | — | 54 |
| 12 | PhIO | DCE | S2 | 80 | — | 52 |
| 13 | PhIO | DCE | S3 | 80 | — | 63 |
| 14 | PhIO | DCE | S4 | 80 | — | 69 |
| 15d | PhIO | DCE | S4 | 80 | — | 73 |
| 16e | PhIO | DCE | S4 | 80 | — | 70 |
| 17f | PhIO | DCE | S4 | 80 | — | 90 |
| 18g | PhIO | DCE | S4 | 80 | — | 82 |
| 19h | PhIO | DCE | S4 | 80 | — | 77 |
| 20i | PhIO | DCE | S4 | 80 | — | 90 |

[a]Reaction conditions: **1a** (0.5 mmol), HIR (1.5 equiv), solvent (5.0 mL), rt. NR = no reaction.
[b]Entries 3–12, additives (1.5 equiv) were used; entries 13–20, additives (0.2 equiv) were used.
[c]Isolated yield.
[d]BF₃•OEt₂ (1.5 equiv) was added.
[e]TMSOTf (1.5 equiv) was added.
[f]LiClO₄ (1.5 equiv) was added.
[g]Zn(ClO₄)₂ (1.5 equiv) was added.
[h]LiClO₄ (1.0 equiv) was added.
[i]LiClO₄ (1.8 equiv) was added. (For details, see Supplementary Table S1 and S2).

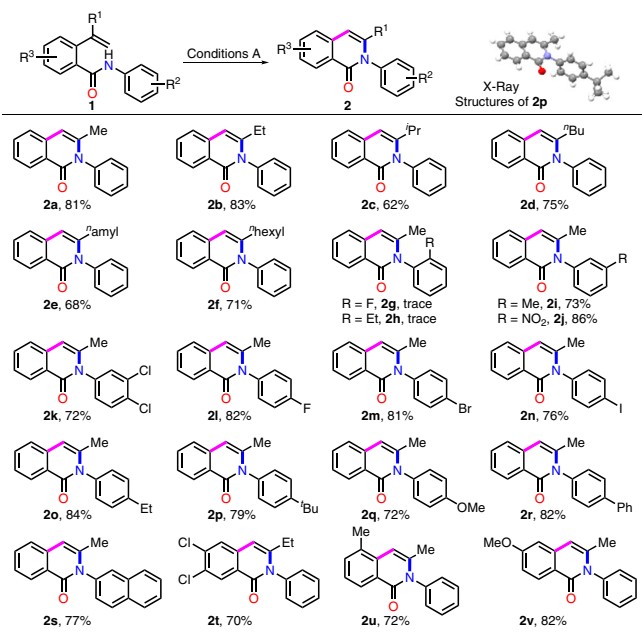

**Fig. 2 Substrate scope study for synthesis of isoquinolinones 2.** [a]
Reaction conditions: **1** (0.5 mmol), PhIO (1.5 equiv) and TMSOTf (0.2 equiv) in MeOH (5.0 mL) reflux for 0.5–2 h. Isolated yield.

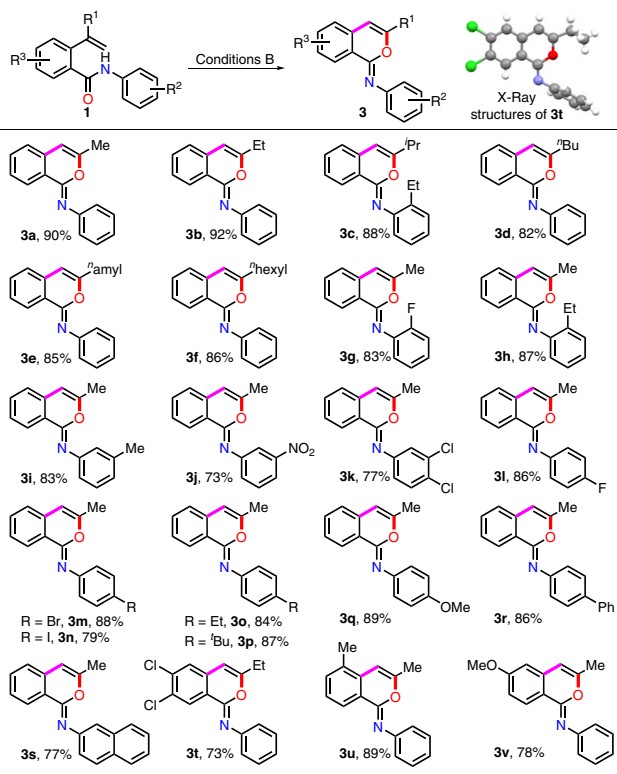

**Fig. 3 Substrate scope study for synthesis of iminoisocoumarins 3.** [a]
Reaction conditions: **1** (0.5 mmol), PhIO (1.5 equiv), 2,4,6-tris-isopropylbenzene sulfonic acid (**S4**; 1.5 equiv) and LiClO₄ (1.5 equiv) in DCE (5.0 mL) at 80 °C for 0.1–0.5 h. Isolated yield.

with other alkyl substituent in the substrate did not have a significant impact on the outcome of the transformation, as substrates **1b-f** were all converted to the desired products **2b-f** with good yields. The method was well applicable to aromatic substituents with either electron-donating groups or electron-withdrawing groups at *meta-* or *para*-position of phenyl ring, furnishing the corresponding isoquinolinones **2i-q** in good yields (70–86%). Furthermore, altering the benzo-ring backbone to biphenyl and naphthyl were also compatible with the reaction conditions, with the corresponding fused heterocycle **2r** and **2s** obtained in 82 and 77% yield respectively. Notably, substrates with electron-poor or electron-rich skeletons (**1t-v**) could be conveniently converted to products **2t-v** in a satisfactory yield, proving that the method could also be applied to substrates with benzamide nucleus bearing substituent. It is worth noting that, for the reaction of substrates **1g-h** under the standard conditions A, the substituent effect of *ortho*-position prevents the cyclization, as only trace amounts of the desired product was produced in each case. To our disappointment, when the methyl group in the substrate was replaced by a phenyl group, the reaction of the corresponding substrate gave a complex mixture and no desired product was achieved under the standard conditions.

Next, we came to explore the chemodivergent synthesis of iminoisocoumarins **3** by subjecting substrates **1** to conditions B (Fig. 3). *O*-alkenylbenzamides **1b–v** bearing different alkyl substituted R¹ group and the substituted phenyl ring could smoothly furnish the corresponding iminoisocoumarins **3b–v** with sole chemoselectivity in satisfactory to excellent yields (73–92%). Notably, in contrast to the inferior performance of substrate **1g** and **1h** in N-cyclization mode reaction, iminoisocoumarin product **3g** and **3h** could be obtained in high yield mediated by the modified Koser's reagent.

In addition, we carried out some control experiments to ascertain the hypervalent iodine species that is responsible for promoting the transformation (Fig. 4). First, we monitored the reaction process of substrate **1a** with PhIO in MeOH in the absence of TMSOTf, and it was found that the reaction did not occur (Fig. 4a). Furthermore, the reaction carried out in the

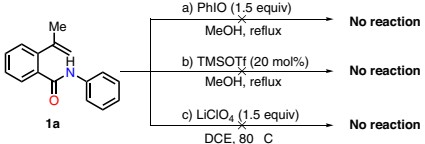

**Fig. 4 Control experiments. a** Control experiment to verify the necessity of hypervalent iodine reagent and Lewis acid. **b** Control experiment to verify the effect of LiClO₄.

absence of PhIO also completely inhibited the transformation of substrate **1a** into product **2a** (Fig. 4b). The two results suggested that both hypervalent iodine reagent and Lewis acid are key reagents that enabled the transformation to occur. Next, when LiClO₄ was solely applied to the transformation of **3a**, no reaction occurred either (Fig. 4c). On the basis of this result as well as the outcome of the initial attempt of using HTIB (Table 1, entry 9), we tentatively infer that LiClO₄ could on one hand promote the formation of Koser's reagent, while on the other hand, coordinate with hydroxy group in the hypervalent iodine specie formed in situ.

To gain insight into the mechanism and chemoselectivity of above systems, we performed density functional theory (DFT) calculations on the reaction of substrate **1a** under conditions A and conditions B (Fig. 5). As previous work shows, PhI(OMe)₂ is generated in situ by PhIO and MeOH[54,56,57]. It is known that TMSOTf can be present as TMS⁺ + TfO⁻ in organic solvents[58,59]. The calculation shows that the complex **IM2-1**, which is formed by PhI(OMe)₂ and TMSOTf, is thermodynamically favored over reagent **1** by 14.0 kcal/mol (Fig. 5a). The amide-iminol tautomerism is achieved via a concerted intermolecular hydrogen exchange between two molecules of substrate **1a** with an

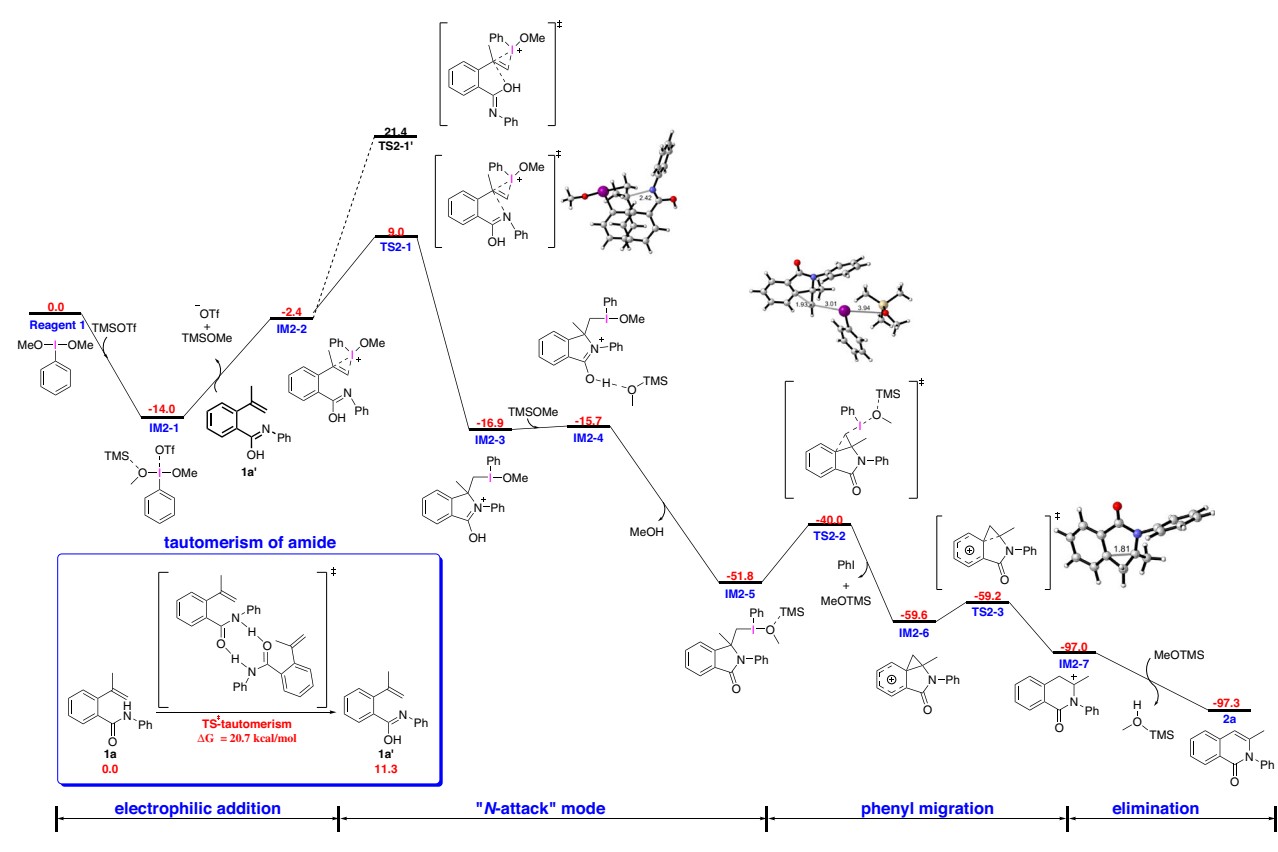

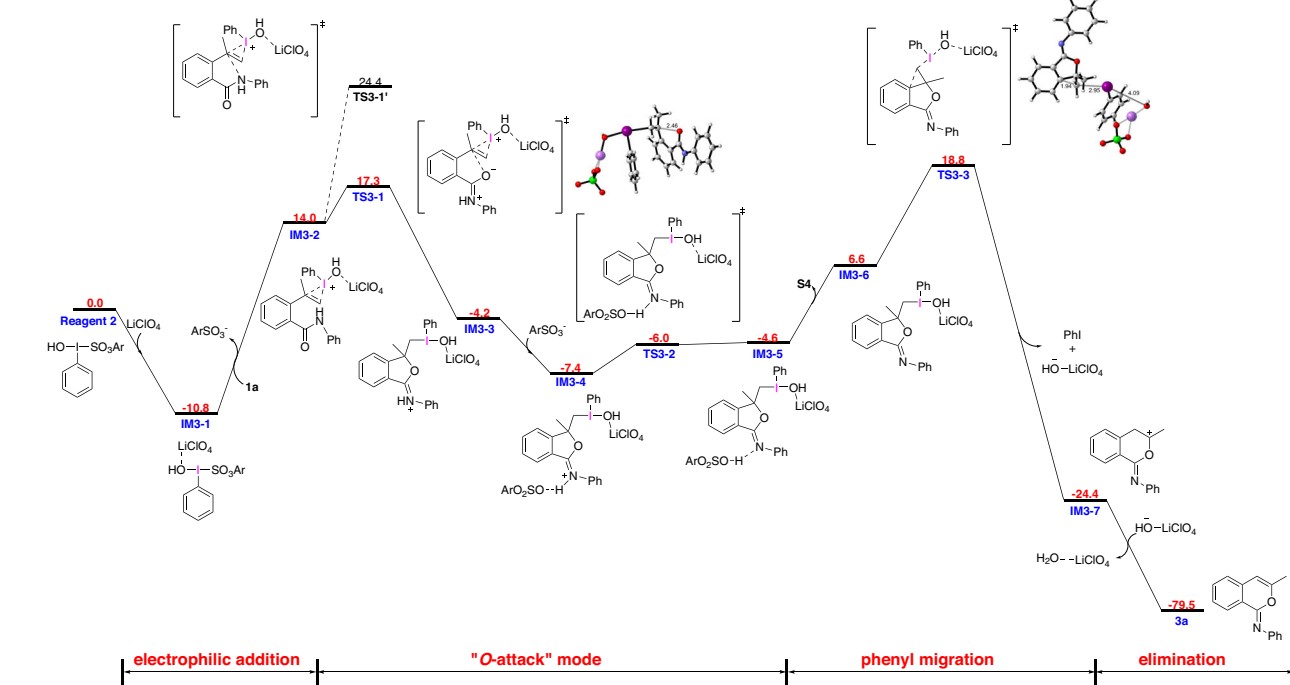

**Fig. 5 DFT-computed potential energy profile for the reaction of 1a under the conditions A and conditions B.** Ar = 2,4,6-triisopropyl (standard state: 25 °C, 1 mol/L). (For details, see Supplementary data 4).

activation energy of 20.7 kcal/mol. Then, the nucleophilic attack onto the iodine(III) center of **IM2-1** by the olefinic double bond of **1a'** proceeds, leading to the formation of iodonium ion **IM2-2**. Then, isomerization of the amide occurs via an intermolecular proton shift with the aid of MeOH as a proton shuttle, which has an activation energy of 26.1 kcal/mol; this process is rate-limiting step in the reaction. Then, the nitrogen atom attacks the more substituted carbon atom on three-membered heterocycle via a 5-exo cyclization in **TS2-1**, which has an energy barrier of 23.0 kcal/mol; this process is the rate-limiting step in the reaction. The nucleophilc attack of oxygen on the three-memebered heterocycle needs to overcome an energy barrier of 35.4 kcal/mol, which is much higher than that required by nitrogen-attack mode. Then, **IM2-4** is generated by the hydrogen bonding interaction between **IM2-3** and MeOTMS. A facile proton shift process occurs to give **IM2-5**. The activated carbon atom bonded to iodine(III)-center is nucleophilically attacked by the phenyl ring, giving a phenonium ion **IM2-6** via **TS2-2**, which has an energy barrier of 11.8 kcal/mol relative to **IM2-5**. Then, ring opening of the three-membered ring in **IM2-6** takes place with simultaneous ring expansion to the six-membered ring **IM2-7** having a tertial carbocation. Finally, a unimolecular elimination process occurs to yield product **2a**. The Gibbs free energy profile of the overall reaction shows that the generation of **2a** is highly exergonic by 97.3 kcal/mol.

For another reaction (Fig. 5b), it starts with the coordination of the oxygen to the LiClO$_4$ which results in the formation of a thermodynamically more stable complex **IM3-1** by 10.8 kcal/mol. Then, the nucleophilic attack onto **IM3-1** by the olefinic double bond of **1a** takes place, which leads to iodonium ion **IM3-2** with the release of ArSO$_3^-$. The isomerization of amide group would not happen as the above reaction, because aprotic solvent 1,2-dichloroethane cannot play a role as proton shuttle. The oxygen of carbonyl attacks the more substituted carbon with an energy barrier of 28.1 kcal/mol, while nitrogen atom accomplishes this process with a higher barrier of 35.2 kcal/mol. After that, **IM3-6** is generated by a proton shift with a barrier of 1.4 kcal/mol relative to **IM3-4**. Subsequently, the phenyl ring migration process occurs with an activation energy of 29.6 kcal/mol, yielding carbocation intermediate **IM3-7**. Finally, product **3a** can be obtained by β- elimination. In this reaction, the rate-limiting step is the phenyl migration process of **IM3-6** with an activation energy of 29.6 kcal/mol. The Gibbs free energy profile of the overall reaction shows that the generation of **3a** is highly exergonic by 79.5 kcal/mol.

Nitrogen or oxygen-attack mode is supported by Hirshfeld charges analysis of **IM2-2** and **IM3-2**, which is capable of predicting electrophilicity and nucleophilicity (Fig. 6). The most important factor responsible for the divergent reactivity of amide moiety is the different nucleophilicity of nitrogen and oxygen atoms under conditions A and B. By the Hirshfeld charges analysis of **IM2-2**, we observed that more negative charge (−0.202) was concentrated on the nitrogen atom than that on the oxygen atom (−0.180), indicating that the nitrogen attacking mode is more favorable for **IM2-2**, thus leading to the formation of N-heterocycle intermediate **IM2-3**. Furthermore, still by the Hirshfeld charges analysis of **IM2-2'** shown below, we found that the negative charge at the oxygen atom of the carbonyl group is -0.364, which means that oxygen has higher nucleophilicity than that of nitrogen of **IM2-2'**, this would result in the generation of O-heterocycle intermediate **IM2-3'**. However, by comparing the Gibbs free energy of **IM2-3** and that of **IM2-3'**, we found that **IM2-3** (−16.9 kcal/mol) is thermodynamically more stable than **IM2-3'** (−14.5 kcal/mol). Thus, N-attack mode is thermodynamically favorable in **IM2-2** under reaction condition A. (Fig. 6a)[60].

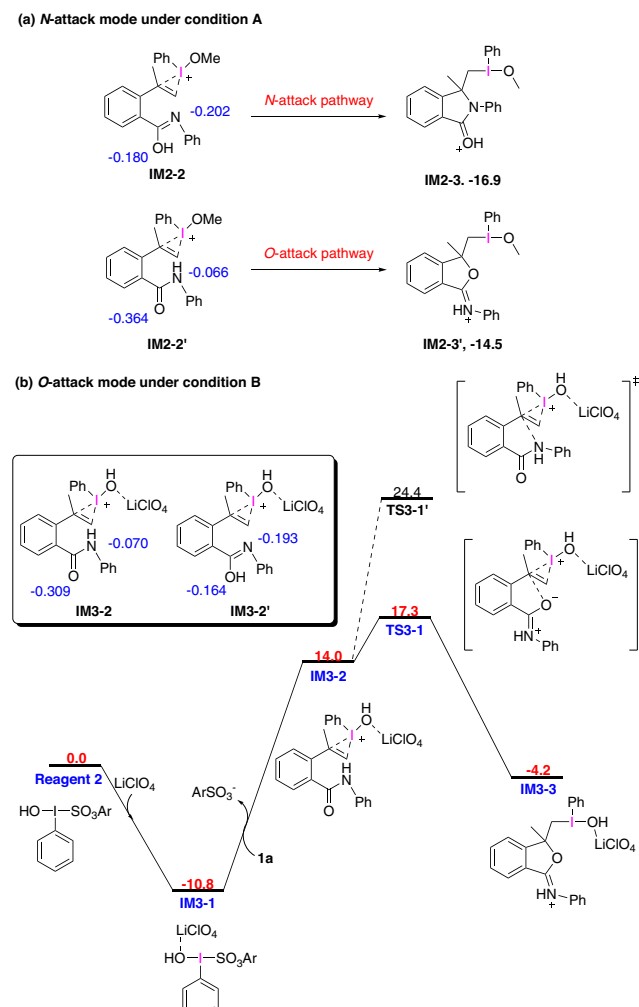

**Fig. 6 The Hirshfeld charges analysis of IM2-3 and IM3-2.** Relative free energies and electronic energies are given in kcal/mol.

Additionally, by the Hirshfeld charges analysis of **IM3-2**, we observed that more negative charge (−0.309) was concentrated on the oxygen atom of amide group than that on the nitrogen atom (−0.070), which indicates that the O-attack mode is more favorable for **IM3-2**. And, we respectively calculated the activation energy barrier for O-attack and N-attack in **IM3-2**. The O-attack pathway needs to overcome an activation energy barrier of 28.1 kcal/mol, while nitrogen atom accomplishes N-attack pathway with a higher barrier of 35.2 kcal/mol. The higher barrier of 35.2 kcal/mol makes it impossible for **IM3-2** to undergo an N-attack mode under condition B. The comparation between Hirshfeld charge of the oxygen atom (−0.309) and that of nitrogen atom (−0.070) also illustrate that O-attack mode is more favorable for **IM3-2**. In summary, O-attack is kinetically favorable under condition B, leading to the generation of O-heterocycle intermediate **IM3-3** under condition B. (Fig. 6b).

Based on the aforementioned mechanistic studies and the outcomes of the previous reports[36,52,54,61,62], we postulated a plausible mechanism for the formation of **2a** and **3a** (Fig. 7). For the formation of product **2a** (Fig. 7a), PhI(OMe)$_2$ is first generated in situ from the reaction of PhIO with MeOH. Then complex **IM2-1** is formed by PhI(OMe)$_2$ and TMSOTf, enabling the electrophilic reaction with isomerized substrate **1a'**, leading to the formation of iodonium ion **IM2-2**. Next, isomerization of the amide occurs via an intermolecular proton shift, and the

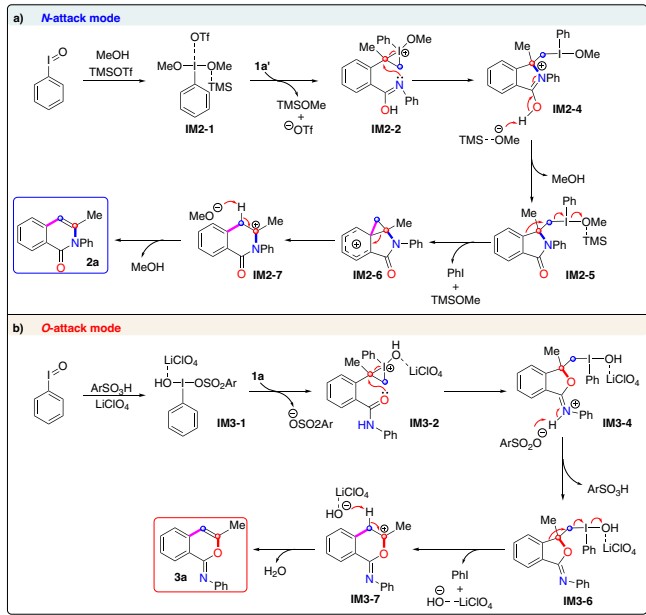

**Fig. 7 Possible reaction mechanism. a** Reaction formation mechanism of isoquinolinone **2a**. **b** Reaction formation mechanism of iminoisocoumarin **3a**.

**Fig. 8 Organocatalytic transformation of chemodivergent synthesis.** Experiment to demonstrate the economic application of this transformation.

subsequent nucleophilic attack of the nitrogen atom gives intermediate **IM2-4**.

Then the deprotonation by the methoxy anion forms intermediate **IM2-5**. The activated carbon atom bonded to iodine-center is nucleophilically attacked by the phenyl ring, giving a phenonium ion **IM2-6**[63–67]. Then, ring opening of the cyclopropane moiety in **IM2-6** occurs with simultaneous ring expansion to give carbocation **IM2-7**. Finally, the elimination reaction occurs in **IM2-7** to form isoquinolinone **2a**. Similarly, with regard to the formation of product **3a** (Fig. 7b), the modified Koser's reagent is first formed from the reaction of PhIO with ArSO₃H. It is worth noting that he coordination of the hydroxyl group to LiClO₄ results in the formation of a thermodynamically more stable complex **IM3-1**, which undergoes electrophilic addition with olefin to form intermediate **IM3-2**. Next, the nucleophilic attack of oxygen atom of the amide moiety onto the benzylic carbon center in **IM3-2** produces intermediate **IM3-4**. After that, deprotonation by the sulfonate ion enables the conversion of iminium **IM3-4** to imine **IM3-6**. Then the aromatic backbone of **IM3-6** undergoes 1,2-aryl migration[36,53,68,69] to form carbon cation **IM3-7**. Finally, with the removal of the acidic proton by the hydroxide anion released from the prior step, intermediate **IM3-7** is converted to iminoisocoumarin **3a** is obtained.

In view of the atomic economy of utilizing catalytic hypervalent iodine species, we further investigated the strategy of combining aryl iodine and exogenous oxidant to generate hypervalent iodine species in situ, with a purpose of demonstrating the economic application of this transformation (Fig. 8)[70–74]. Gratefully, after screening the conditions of the catalytic conversion, we

found that the PhI-catalyzed reaction of substrate **1a** occurred smoothly by using *m*CPBA (*meta*-chloroperoxybenzoic acid) as the terminal oxidant, combined with acetic acid and stoichiometric amount of TMSOTf in MeOH at reflux temperature, with isoquinolinone **2a** achieved in a moderate 56% yield (Fig. 8a). Furthermore, iminoisocoumarin **3a** could be obtained in 92% yield when substrate **1a** was treated with catalytic amount of PhI (20 mol%), *m*CPBA (2.0 equiv) in the presence of 2,4,6-tris-iso-propylbenzene sulfonic acid (**S4**; 1.5 equiv) and LiClO₄ (0.2 equiv) in DCE at 80 °C (Fig. 8b).

In summary, we have presented an exclusive chemodivergent cycloisomerization approach for constructing isoquinolinones and iminoisocoumarins skeletons starting from an identical *o*-alkenylbenzamides derivative. Notably, the divergent synthesis employed an exclusive 1,2-aryl migration/elimination strategy, which is realized by utilizing the different hypervalent iodine species generated in situ. In the reaction processes, different hypervalent iodine species was found to play a crucial role in the selectivity switch from N to O-cyclization, with the PhI(OMe)₂ species inducing N-attack and the modified Koser's reagent favoring the O-attack in the cyclization step. DFT studies demonstrated that nitrogen and oxygen atoms of the intermediates in the two reaction systems have different nucleophilicity and thus produce the selectivity of N or O-attack mode.

## Methods

**General procedure for synthesis of isoquinolinones 2**. To a reaction flask filled with iodosobenzene (1.5 equiv, 0.75 mmol) in MeOH (5.0 mL) was added TMSOTf (20 mol %). The mixture was stirred at reflux temperature for 5 min and then reactant **1** (0.5 mmol) was added. The resulting mixture was kept stirring until TLC indicated the total consumption of substrate **1**. Then the reaction mixture was quenched with sat. aq. NaHCO₃ (5 mL), and extracted with EtOAc (10 mL × 3). The combined organic layer was dried over anhydrous Na₂SO₄ and the solvent was removed in vacuo. The residue was purified by flash column chromatography on silica gel (petroleum ether/ethyl acetate 10:1) to afford target product **2**. Details including experimental procedures are available in the Supplementary Methods (Supplementary Table S1).

**General procedure for synthesis of iminoisocoumarins 3**. To a reaction flask filled with iodosobenzene (1.5 equiv, 0.75 mmol) in DCE (5.0 mL) was added **S4** (1.5 equiv, 0.75 mmol) and LiClO₄ (1.5 equiv, 0.75 mmol). The mixture was stirred at 80 °C for 5 min and then reactant **1** (0.5 mmol) was added. The resulting mixture was kept stirring until TLC indicated the total consumption of substrate **1**. Then the mixture was quenched with sat. aq. NaHCO₃ (5 mL), and extracted with dichloromethane (10 mL × 3). The combined organic layer was dried over anhydrous Na₂SO₄ and the solvent was removed in vacuo. The residue was purified by flash column chromatography on silica gel (petroleum ether/ethyl acetate 50:1) to afford target product **3**. Details including experimental procedures are available in the Supplementary Methods (Supplementary Table S2).

## Data availability

All data generated during this study are included in this article and Supplementary Information. Experimental procedure, conditions optimization and product characterization are provided in the Supplementary Information. The NMR spectra of all compounds are available in Supplementary Data 1. The X-ray crystallographic coordinates for structures reported in this Article have been deposited at the Cambridge Crystallographic Data Centre (CCDC), under deposition number 2201660 (**2p**, Supplementary Tables S3–S9, Supplementary Data 2) and 2202945 (**3t**, Supplementary Table S10–S17, Supplementary Data 3), respectively. These data can be obtained free of charge from the Cambridge Crystallographic Data Centre via www.ccdc.cam.ac.uk/data_request/cif. DFT calculations are available in Supplementary Data 4.

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

## Acknowledgements

Y.D. acknowledges the National Natural Science Foundation of China (No. 22071175). We also thank Dr. Jun Xu, Dr. Yan Gao, and Professor Xiangyang Zhang [AIC, SPST/TJU] for providing the analysis support.

## Author contributions

J.H. and Y.D. conceived and designed the experiments. J.H. performed all the experiments and prepared the manuscript and supporting information. F.-H.D. performed all the DFT calculations work and collated the calculated data. C.Z. and Y.D. directed the research and revised the manuscript.

## Competing interests

The authors declare no competing interests.
