## [Peer Review File · Communications Chemistry]

Reviewers' comments:

Reviewer #1 (Remarks to the Author):

My primary role in reviewing this manuscript was confirming that veracity of the iodine reagent used. I have to admit I was skeptical of $\text{PhI}(\text{OMe})_2$, but was pleasantly surprised to find out a crystal structure has been reported (the 2021 Chem Eur J properly cited here). I cannot comment if the transformations presented here are of interest to organic chemists or anyone, or if they are amazing. I leave that to the other referees and the editor. My eye is this is publishable as far as my expertise goes.

The DFT calculations also appear to be good. The proposed mechanisms are...complicated and who knows if this is how the chemistry actually goes. But the energy barriers (~ 100 kJ/mol) are at least consistent with the reaction conditions (reflux for a little while in MeOH or DCE) which is better than a lot of papers manage.

My one comment is the assertion about TMS-OTf. In DCE this is certainly intact as TMS-OTf and not cation/anion. It is interesting their first step of splitting TMS-OTf on the Me-O-I fragment, and that does seem energetically favourable.

Reviewer #2 (Remarks to the Author):

This article by Du & Zhang and coworkers titled "Concomitant 1,2-Aryl Migration/Elimination Reaction Mediated by Hypervalent Iodine Reagents: Regioselective Cycloisomerization of O-Alkenylbenzamides" offers an interesting discovery in HVI-mediated heterocycle synthesis. The substrate scope for both cyclizations was more than adequate, and the product characterization (including Xray crystallography) have more than convinced me of the product identities. I have thoroughly examined the SI document spectra and they are predominantly clean and clear of impurities. The characterization data is also very well compiled, and of the 10 random examples that I checked against their spectra (H/C counts, peak labels and J values), I found no errors. Similarly, the interpretation of fluorine splitting in the NMRs was very well done, which is commendable.

The writing in the manuscript is generally well done, but I have made notations on the first couple pages (see attachment) which could be addressed and applied to the remainder of the document as well.

There are several areas that I would like to see addressed:

1. If the products contain different functional groups, is it not chemoselectivity (rather than regioselectivity) that is being observed? This could warrant a change in title/abstract.
2. The computational results have me asking many questions. In terms of presentation, why is the 1,2-phenyl shift explicitly drawn out in 1a but completely overlooked in 1b? Along these lines, why is a deprotonation shown so explicitly (IM3-4 to IM3-6) at the expense (in terms of page space) of this phenyl shift. Are the authors trying to show the rationale for this deprotonation being 14 kcal/mol uphill? (This seems like a dramatic increase in energy for deprotonating an iminium ion, does it not? Even in these conditions.)

3. For the N-cyclization pathway, I am struggling with the idea that a proton transfer (via TS2-1) is the rate limiting step. Could the reaction not be proceeding via the small amount of the amide 'enol' already present? Why does this have to tautomerize after the iodoiranium ion has formed, and not before. Can this intermediate be calculated and presented?

4. I am also trying to understand why the HVI reagent would engage the weakly Lewis basic alkene when the more nucleophilic amide oxygen is available? In my mind, I am envisioning a process that proceeds via an activated amide (similar to the Szpilman enolonium ions), but to be truthful, I tried to draw this and couldn't come up with something reasonable for the required 1,2-phenyl shift.

Perhaps these other pathways (comments 2-4) could also be investigated and eliminated from consideration due to the resulting energy barriers. I am not asking for more evidence that the authors are correct in their mechanistic proposal, rather, I am suggesting they include a justification for why other (seemingly plausible) proposals are not.

5. The Hirshfeld charges shown in Figure 2 are informative, but incomplete in my assessment. Rather than simply showing the charges for the intermediates proposed, the authors should also show the charges for the other 'enol' forms for each of the intermediates. This would allow for a more comprehensive comparison of the charges, and better support the claims of why the reaction proceeds this way.

6. I am rather amazed that the catalytic process for forming 3a actually works better than the stoichiometric $\text{PhI}=\text{O}$ reaction. Especially since LiClO_4 loading has been decreased to 20 mol% (from 1.5 equiv), and that there are so many oxygenated byproducts floating around in solution in this reaction (eg. mCBA).

7. In the mechanism of Scheme 3, the arrow on intermediate D is misdirected. The arene appears to be attacking the nitrogenated carbon, rather than that with the iodoarene leaving group.

In summary, I like this work. It has been very well done and it's of excellent experimental quality. The computational results have me asking several questions that I can't seem to rationalize, and so I feel others might have the same questions if left as-is. And given that the differing chemoselectivity is the major selling point of this article, a clearer explanation of 'why' is likely appropriate. Should the authors include additional info to address some of these concerns, I believe this would be an excellent publication for CommsChem

Reviewer #3 (Remarks to the Author):

The key point of the calculation is to understand the selectivity of the N/O attack. Under conditions A, the two competing transition states are $-\text{OH}/=\text{N}-$ attacking; However, Under conditions B, the two competing transition states are $-\text{O}/-\text{NH}-$ attacking. Under conditions A, the methanol is used as a proton

shuttle. Under conditions B, ArSO₃H can also do this. Therefore, I think, for both conditions A and B, four types transition states need to be calculated: -O/OH and =N/=NH attacking.

Part I: Reviewer's suggestions

Reviewer 1's comments:

My primary role in reviewing this manuscript was confirming that veracity of the iodine reagent used. I have to admit I was skeptical of $\text{PhI}(\text{OMe})_2$, but was pleasantly surprised to find out a crystal structure has been reported (the 2021 Chem Eur J properly cited here). I cannot comment if the transformations presented here are of interest to organic chemists or anyone, or if they are amazing. I leave that to the other referees and the editor. My eye is this is publishable as far as my expertise goes.

The DFT calculations also appear to be good. The proposed mechanisms are...complicated and who knows if this is how the chemistry actually goes. But the energy barriers (~ 100 kJ/mol) are at least consistent with the reaction conditions (reflux for a little while in MeOH or DCE) which is better than a lot of papers manage.

My one comment is the assertion about TMS-OTf. In DCE this is certainly intact as TMS-OTf and not cation/anion. It is interesting their first step of splitting TMS-OTf on the Me-O-I fragment, and that does seem energetically favourable.

Response: We highly appreciate the professional and constructive comment and suggestion from this Reviewer. In the intermediate **IM2-1** of Figure 1a, the bond length of Si-O is 1.73 Å, and the bond length of I(III)-OTf is 2.88 Å. According to a literature report (*Inorganic Materials*, **2001**, 37, 871), the van der Waals radii of the atom of Si, O, and I is 2.1 Å, 1.55 Å, and 2.1 Å, respectively. In **IM2-1**, the bond length of Si-O bond (1.73 Å) and I-O bond (2.88 Å) are shorter than the sum of the van der Waals radii of either Si and O or I and O (both 3.65 Å), which illustrates that there is a bonding interaction between Si and O, and a secondary bonding between I and O. These two interactions result in the decreasing free energy of **IM2-1** compared with that of $\text{PhI}(\text{OMe})_2$, thus leading to the dissociation of TMSOTf in the reaction.

IM2-1

Reviewer 2's comments:

This article by Du & Zhang and coworkers titled “Concomitant 1,2-Aryl Migration/Elimination Reaction Mediated by Hypervalent Iodine Reagents: Regioselective Cycloisomerization of *O*-Alkenylbenzamides” offers an interesting discovery in HVI-mediated heterocycle synthesis. The substrate scope for both cyclizations was more than adequate, and the product characterization (including X-ray crystallography) have more than convinced me of the product identities. I have thoroughly examined the SI document spectra and they are predominantly clean and clear of impurities. The characterization data is also very well compiled, and of the 10 random examples that I checked against their spectra (H/C counts, peak labels and J values), I found no errors. Similarly, the interpretation of fluorine splitting in the NMRs was very well done, which is commendable.

The writing in the manuscript is generally well done, but I have made notations on the first couple pages (see attachment) which could be addressed and applied to the remainder of the document as well.

Response: We thank the Reviewer for putting forward this helpful suggestion. We have got it corrected in the revised manuscript. Please find the detailed changes in the Table afterwards.

There are several areas that I would like to see addressed:

1. If the products contain different functional groups, is it not chemoselectivity (rather than regioselectivity) that is being observed? This could warrant a change in title/abstract.

Response: We thank this reviewer for this professional and constructive suggestion. We have checked and got it corrected in the revised manuscript. Please find the detailed changes in the Table afterwards.

2. The computational results have me asking many questions. In terms of presentation, why is the 1,2-phenyl shift explicitly drawn out in **1a** but completely overlooked in **1b**? Along these lines, why is a deprotonation shown so explicitly (**IM3-4** to **IM3-6**) at the expense (in terms of page space) of this phenyl shift. Are the authors trying to show the rationale for this deprotonation being 14 kcal/mol uphill? (This seems like a dramatic increase in energy for deprotonating an iminium ion, does it not? Even in these conditions.)

Response: We thank the Reviewer for the professional comment. To answer the question, we performed the intrinsic reaction coordinate (IRC) calculation on **TS3-3** which is shown in the figure below. Generally, IRC calculation defines the validity of transition states and is used to search the structures of reactants and products connecting to the transition states. In the figure below, the structure of **TS3-3** is set to the horizontal coordinate origin, the negative and positive regions of the IRC are defined to correspond to reactant region and product region, respectively. As the IRC calculation shows, the reactant of **TS3-3** is **IM3-6**, and the product of the same transition state is carbocationic species **IM3-7**. Thus, the description of 1,2-phenyl shift is reasonable in Figure 1b of our submitted manuscript. The very similar 1,2-phenyl shift process were found in literatures, please see the references: (1) *J. Org. Chem.* **2016**, *81*, 4058; (2) Figure 5, *ACS Catal.* **2017**, *7*, 1093.

Just as the Reviewer 2 stated, the reason for the detailed calculations of the deprotonation process from **IM3-4** to **IM3-6** is to show the rationale for the uphill 14 kcal/mol energy barrier. It is not a thermodynamically favored process as usual work (please see the references: *Angew. Chem. Int. Ed.* **2021**, *60*, 24171; and *J. Org. Chem.*, **2016**, *81*, 9006) because a strong acid (**S4**) is generated from a weaker acid (**IM3-3**). Therefore, we explicitly presented the deprotonation of the iminium ion **IM3-3** for better understanding.

3. For the *N*-cyclization pathway, I am struggling with the idea that a proton transfer (via **TS2-1**) is the rate limiting step. Could the reaction not be proceeding via the small amount of the amide 'enol' already present? Why does this have to tautomerize after the iodoiranium ion has formed, and not before. Can this intermediate be calculated and presented?

Response: As requested by Reviewer 2, we calculated the energy of the tautomerism process shown in the Figure 1 below. The black line represents the original mechanism, while the blue line indicates the revised one. As the Figure 1 shows, the amide-iminol tautomerism is achieved via a concerted intermolecular hydrogen exchange process between two molecules of substrate **1a** with an activation energy of 20.7 kcal/mol. In

contrast, iodonium ion requires an activation energy of 26.1 kcal/mol to achieve the amide-iminol tautomerism. Therefore, it is energetically favorable for the tautomerism process to take place in **1a** before the formation of iodonium ion.

Figure 1. DFT-computed potential energy profile for the tautomerism of **1a**.

On the basis of new calculation results, the revised potential energy profile for N-cyclization pathway is shown in the Figure 2. The tautomerism of **1a** occurs before the formation of iodonium ion. The rate-limiting step is the nucleophilic attack of nitrogen atom with an activation energy of 23.0 kcal/mol. The following revised potential energy profile and the corresponding description have been put in the revised manuscript.

Figure 2. DFT-computed potential energy profile for the reaction of **1a** under the conditions A.

4. I am also trying to understand why the HVI reagent would engage the weakly Lewis basic alkene when the more nucleophilic amide oxygen is available? In my mind, I am envisioning a process that proceeds via an activated amide (similar to the Szpilman enolonium ions), but to be truthful, I tried to draw this and couldn't come up with something reasonable for the required 1,2-phenyl shift.

Response: As advised by reviewer 2, we calculated the intermediate **A** formed by the combination of $\text{PhI}(\text{OMe})_2$ and the amide group, and found that this intermediate is more stable than the iodonium ion **IM2-2** by 6.1 kcal/mol. However, the attack onto the oxygen atom bonded to iodine(III)-center of **A** by the carbon atoms either at the benzyl position or β position of the C-C double bond need to overcome 82.0 kcal/mol and 46.1 kcal/mol activation energy barriers, respectively. So, both pathways cannot take place under the condition A. Consequently, the hypervalent iodine(III) intermediate **IM2-1** generated *in situ* would prefer engaging the C-C double bond of **1a**. This activation mode can also be found in literatures (Please see the two references: *J. Org. Chem.* **2016**, *81*, 9006; *J. Org. Chem.* **2019**, *84*, 15605).

5. The Hirshfeld charges shown in Figure 2 are informative, but incomplete in my assessment. Rather than simply showing the charges for the intermediates proposed, the authors should also show the charges for the other ‘enol’ forms for each of the intermediates. This would allow for a more comprehensive comparison of the charges, and better support the claims of why the reaction proceeds this way.

Response: By the Hirshfeld charges analysis of **IM2-2**, we observed that more negative charge (-0.202) was concentrated on the nitrogen atom than that on the oxygen atom (-0.180), indicating that the nitrogen attacking mode is more favorable for **IM2-2**, thus leading to the formation of N-heterocycle intermediate **IM2-3**. Furthermore, still by the Hirshfeld charges analysis of **IM2-2'** shown below, we found that the negative charge at the oxygen atom of the carbonyl group is -0.364, which means that oxygen has higher nucleophilicity than that of nitrogen of **IM2-2'**, this would result in the generation of O-heterocycle intermediate **IM2-3'**. However, by comparing the Gibbs free energy of **IM2-3** and that of **IM2-3'**, we found that **IM2-3** (-16.9 kcal/mol) is thermodynamically more stable than **IM2-3'** (-14.5 kcal/mol). Thus, N-attack mode is thermodynamically favorable in **IM2-2** under reaction condition A.

By the Hirshfeld charges analysis of **IM3-2**, we observed that more negative charge (-0.309) was concentrated on the oxygen atom of amide group than that on the nitrogen atom (-0.070), which indicates that the O-attack mode is more favorable for **IM3-2**. And, we respectively calculated the activation energy barrier for O-attack and N-attack in **IM3-2**. The O-attack pathway needs to overcome an activation energy barrier of 28.1

kcal/mol, while nitrogen atom accomplishes *N*-attack pathway with a higher barrier of 35.2 kcal/mol. The higher barrier of 35.2 kcal/mol makes it impossible for **IM3-2** to undergo an *N*-attack mode under condition B. The comparison between Hirshfeld charge of the oxygen atom (-0.309) and that of nitrogen atom (-0.070) also illustrate that *O*-attack mode is more favorable for **IM3-2**. In summary, *O*-attack is kinetically favorable under condition B, leading to the generation of *O*-heterocycle intermediate **IM3-3** under condition B.

6. I am rather amazed that the catalytic process for forming **3a** actually works better

than the stoichiometric $\text{PhI}=\text{O}$ reaction. Especially since LiClO_4 loading has been decreased to 20 mol% (from 1.5 equiv), and that there are so many oxygenated byproducts floating around in solution in this reaction (eg. *mCBA*).

Response: We thank the Reviewer for this professional question. To our understanding, the formation of product **3a** was very rapid under the reaction conditions, and some side reactions inevitably occurred. In this context, the reaction might be better facilitated by the addition of LiClO_4 in equivalent ratios for the case when stoichiometric amounts of hypervalent iodine reagent were applied. However, in catalytic reaction process, the hypervalent iodine specie could only be generated in situ when catalytic iodobenzene underwent oxidation first, and then reacted with substrate **1a**. Thus adding a small amount of LiClO_4 , in conjunction with the catalytic amount of the formed hypervalent iodine reagent, might efficiently promote the formation of **3a**. Furthermore, we postulated that the formed catalytic amount of hypervalent iodine species could react with substrate **1a** slowly, which greatly inhibit the undesired side reactions and thus improve the yield of the desired product.

In addition, during our monitor of the reaction process for the formation of **3a**, we observed that the absence of sulfonic acid resulted in the complete inhibition of this transformation. But when sulfonic acid was added, **3a** was instantly formed. This result clearly indicates that the reaction did not occur in the absence of sulfonic acid. Furthermore, when sulfonic acid was replaced with *mCBA*, we found that the reaction did not occur either. This result showed that byproduct such as *mCBA* does not influence the formation of **3a**. With these regards, we tentatively postulate that in the reaction system, LiClO_4 preferentially involved in the reaction process to facilitate the formation of product **3a**, rather than complexed with *mCBA*.

7. In the mechanism of Scheme 3, the arrow on intermediate **D** is misdirected. The arene appears to be attacking the nitrogenated carbon, rather than that with the iodoarene leaving group.

Response: We are sorry for this drawing mistake. We have corrected it in the revised manuscript. Please find the detailed changes in the Table afterwards.

Reviewer 3's comments:

The key point of the calculation is to understand the selectivity of the N/O attack. Under conditions A, the two competing transition states are -OH/=N- attacking; However, under conditions B, the two competing transition states are -O/-NH- attacking. Under conditions A, the methanol is used as a proton shuttle. Under conditions B, ArSO₃H can also do this. Therefore, I think, for both conditions A and B, four types transition states need to be calculated: -O/OH and =N/=NH attacking.

Response: The questions raised by Reviewer 3 overlap to some extent with those raised by Reviewer 2. Following the comments and suggestions raised by Reviewers 1 and 2, we did calculation works and had the mechanism revised for the reaction going under the condition A. Particularly, to answer the 5th question of Reviewer 2 (related to the Hirshfeld charges analysis), the answer we obtained is that the nitrogen-attack mode is advantageous under condition A, and the oxygen-attack mode is preferred under the condition B.

Table. A Point-to-point Description of Changes

Location	Original MS	Revised MS
Page 1 Title	Concomitant 1,2-Aryl Migration/Elimination Reaction Mediated by Hypervalent Iodine Reagents: Regioselective Cycloisomerization of O - Alkenylbenzamides	Concomitant 1,2-Aryl Migration/Elimination Reaction Mediated by Hypervalent Iodine Reagents: Chemoselective Cycloisomerization of O - Alkenylbenzamides
Page 1 Abstract	A regiodivergent cycloisomerization approach to construct isoquinolinone and iminoisocoumarin skeletons from o -alkenylbenzamide derivatives is established. The regio-controllable strategy employed an exclusive 1,2-aryl migration/elimination cascade, enabled by different hypervalent iodine species generated in situ from	A chemodivergent cycloisomerization approach to construct isoquinolinone and iminoisocoumarin skeletons from o -alkenylbenzamide derivatives is established. The chemo -controllable strategy employed an exclusive 1,2-aryl migration/elimination cascade, enabled by different hypervalent iodine species generated in situ from the reaction of

	the reaction of iodosobenzene (PhIO) with MeOH or 2,4,6-tris-isopropylbenzene sulfonic acid. DFT studies revealed that the nitrogen and oxygen atoms of the intermediates in the two reaction systems have different nucleophilicity and thus produce the selectivity of N or O-attack modes.	iodosobenzene (PhIO) with MeOH or 2,4,6-tris-isopropylbenzene sulfonic acid. DFT studies revealed that the nitrogen and oxygen atoms of the intermediates in the two reaction systems have different nucleophilicities and thus produce the selectivity of N or O-attack modes.
Page 1 Left column Paragraph 1	Generally, the reaction chemoselectivities of the ambident amide compounds can be controlled by the utilization of appropriate catalysts and reagents.	Generally, the reaction chemoselectivities of ambident amide compounds can be controlled by the utilization of appropriate catalysts and reagents.
Page 1 Left column Paragraph 1	For examples, Nishikata's group has reported an intermolecular cyclization of α-bromoamides and acrylates, with the reactivity of the nitrogen or oxygen nucleophile of the amide group being finely controlled by using a copper catalyst system with an appropriate base (Scheme 1a).	For examples, Nishikata's group has reported an intermolecular cyclization of α-bromoamides and acrylates, with the reactivity of the nitrogen or oxygen nucleophile of the amide group being finely controlled by using a copper catalyst with an appropriate base (Scheme 1a).
Page 1 Left column Paragraph 1	Belmont and coworkers demonstrated the divergent synthesis of five-membered heterocyclic isoindolinones and isobenzofuranones by silver-catalyzed cycloisomerization reaction.	Belmont and coworkers demonstrated the divergent synthesis of five-membered heterocyclic isoindolinones and isobenzofuranones by silver-catalyzed cycloisomerization.
Page 1 Left column Paragraph 1	However, differing from Belmont's work, Jiang and colleagues realized the divergent synthesis of six-membered isoquinolinone and iminoisocoumarin derivatives, the reaction of which was controlled by using gold/ligand and platinum catalyst (Scheme 1b).	However, differing from Belmont's work, Jiang and colleagues realized the divergent synthesis of six-membered isoquinolinone and iminoisocoumarin derivatives, the reaction of which was controlled by using gold/ligand or platinum catalyst (Scheme 1b).
Page 1 Left column Paragraph 1	In this regard, a new protocol dictating the regiodivergent cycloisomerization of o-alkenylbenzamides.	In this regard, a new protocol dictating the chemodivergent cycloisomerization of o-alkenylbenzamides.

Scheme 1		Page 1 Right column Paragraph 1	It should be noted that there are no precedents on hypervalent iodine species controlled chemoselectivity in the cyclization of o-alkenylbenzamide, and the work described here highlights a one-pot transformation involving an exclusive cascade sequences of regioselective cyclization, 1,2-aryl migration and elimination processes.	It should be noted that there are no precedents on hypervalent iodine controlled chemoselectivity in the cyclization of o-alkenylbenzamides, and the work described here highlights a one-pot transformation involving an exclusive cascade sequence of chemoselective cyclization, 1,2-aryl migration and elimination processes.
Page 1 Right column Paragraph 2	After screening the effect of the activators, we found that TMSOTf was the most efficient catalyst for this transformation (Table 1, entry 5).	After screening the effect acidic or basic of activators (entries 2-7), we found that TMSOTf was the most efficient catalyst for this transformation (Table 1, entry 5).
Page 2 Left column Paragraph 3	Encouraged by the above results, we then turned our attention toward exploring the regiodivergent pattern of the protocol for the synthesis of O-cyclized iminoisocoumarin products.	Encouraged by the above results, we then turned our attention toward exploring the chemodivergent pattern of the protocol for the synthesis of O-cyclized iminoisocoumarin products.
Page 2 Left column Paragraph 3	First, a detailed screening of solvents, hypervalent iodine reagents and temperature was carried out (see the SI for details).	First, a detailed screening of solvents, hypervalent iodine reagents and temperature was carried out (see the SI for complete details).
Page 2 Right column Paragraph 3	Next, the utilization of the active hypervalent iodine species, formed in situ from the reaction of iodosobenzene (PhIO) and 4-toluene sulfonic acid,^{51, 52} gave almost the identical result (Table 1, entry 11).	Next, almost identical results were obtained when using active hypervalent iodine species formed in situ from iodosobenzene (PhIO) and 4-toluenesulfonic acid (Table 1, entry 11).^{51, 52}
Page 2 Right column Paragraph 3	Furthermore, the sulfonic acids with different structures were further tested (Table 1, entries 12-14).	Furthermore, sulfonic acids with different structures were further tested (Table 1, entries 12-14).
Page 2 Right column	To our delights, better yields were further achieved with the addition of	To our delight, better yields were further achieved with the addition of an

Paragraph 3	an exogenous Lewis acid (Table 1, entries 15-18).	exogenous Lewis acid (Table 1, entries 15-18).
Page 2 Right column Paragraph 3	Further investigation on the dosage of LiClO ₄ indicated that neither decreasing nor increasing the equivalent would be beneficial for the outcome of the yielding (Table 1, entries 19-20).	Further investigation on the loading of LiClO ₄ indicated that neither decreasing nor increasing the equivalents were beneficial (Table 1, entries 19-20).
Page 2 Right column Paragraph 4	With the optimized conditions in hand, we began to explore the general applicability of the divergent transformation, targeting with the N -cyclization for accessing isoquinolinones by using variously substituted N -phenyl-2-alkenylbenzamides as substrates and MeOH as a reaction partner ^{34, 53-57} and solvent (Table 2) being first studied.	With the optimized conditions in hand, we began to explore the general applicability of the divergent transformation, targeting the N -cyclization by using variously substituted N -phenyl-2-alkenylbenzamides with MeOH as a reaction partner ^{34, 53-57} and solvent (Table 2) being first studied.
Page 3 Left column Paragraph 5	Next, we came to explore the regiodivergent synthesis of iminoisocoumarins 3 by subjecting substrates 1 to conditions B. As shown in Table 3, o -alkenylbenzamides 1b-v bearing different alkyl substituted R1 group and the substituted phenyl ring could smoothly furnish the corresponding iminoisocoumarins 3b-v with sole regioselectivity in satisfactory to excellent yields (73–92%).	Next, we came to explore the chemodivergent synthesis of iminoisocoumarins 3 by subjecting substrates 1 to conditions B. As shown in Table 3, o -alkenylbenzamides 1b-v bearing different alkyl substituted R1 group and the substituted phenyl ring could smoothly furnish the corresponding iminoisocoumarins 3b-v with sole chemoselectivity in satisfactory to excellent yields (73–92%).
Page 3 Right column Paragraph 7	Added	The amide-iminol tautomerism is achieved via a concerted intermolecular hydrogen exchange between two molecules of substrate 1a with an activation energy of 20.7 kcal/mol.
Page 3 Right column Paragraph 7	Then, the nucleophilic attack onto the iodine(III) center of IM2-1 by the olefinic double bond of 1a proceeds, leading to the formation of iodonium ion IM2-2 .	Then, the nucleophilic attack onto the iodine(III) center of IM2-1 by the olefinic double bond of 1a' proceeds, leading to the formation of iodonium ion IM2-2 .
Page 3 Right column Paragraph 7	Then, the nitrogen atom attacks the more substituted carbon atom on three-membered heterocycle via a 5-	Then, the nitrogen atom attacks the more substituted carbon atom on three-membered heterocycle via a 5-exo cyclization in TS2-1 , which has an

	exo cyclization in TS2-2 , which has an energy barrier of 23.0 kcal/mol.	energy barrier of 23.0 kcal/mol; this process is the rate-limiting step in the reaction.
Page 3 Right column Paragraph 7	Then, IM2-5 is generated by the hydrogen bonding interaction between IM2-4 and MeOTMS. A facile proton shift process occurs to give IM2-6 . The activated carbon atom bonded to iodine(III)-center is nucleophilically attacked by the phenyl ring, giving a phenonium ion IM2-7 via TS2-3 , which has an energy barrier of 11.8 kcal/mol relative to IM2-6 . Then, ring opening of the three-membered ring in IM2-7 takes place with simultaneous ring expansion to the six-membered ring IM2-8 having a tertial carbocation.	Then, IM2-4 is generated by the hydrogen bonding interaction between IM2-3 and MeOTMS. A facile proton shift process occurs to give IM2-5 . The activated carbon atom bonded to iodine(III)-center is nucleophilically attacked by the phenyl ring, giving a phenonium ion IM2-6 via TS2-2 , which has an energy barrier of 11.8 kcal/mol relative to IM2-5 . Then, ring opening of the three-membered ring in IM2-6 takes place with simultaneous ring expansion to the six-membered ring IM2-7 having a tertial carbocation.
Page 4 Figure 1		Page 5 Left column Paragraph 9	Nitrogen or oxygen-attack mode is supported by Hirshfeld charges analysis of IM2-3 and IM3-2 , which is capable of predicting electrophilicity and nucleophilicity. The most important factor responsible for the divergent reactivity of amide moiety is the different nucleophilicity of nitrogen and oxygen atoms under conditions A and B. In IM2-3 , the negative charge at the nitrogen atom is -0.202, while oxygen atom of hydroxyl group is -0.180, indicating that nitrogen atom has higher nucleophilicity than oxygen atom, therefore, leading to the formation of N -heterocycle IM2-4 favorably (Figure 2a). ⁶⁰ Additionally, the Hirshfeld charges of the oxygen atom of carbonyl is -0.309 bigger than that of nitrogen atom in IM3-2 , leading to the formation of O -heterocycle IM3-3 (Figure 2b).	Nitrogen or oxygen-attack mode is supported by Hirshfeld charges analysis of IM2-2 and IM3-2 , which is capable of predicting electrophilicity and nucleophilicity. The most important factor responsible for the divergent reactivity of amide moiety is the different nucleophilicity of nitrogen and oxygen atoms under conditions A and B. By the Hirshfeld charges analysis of IM2-2, we observed that more negative charge (-0.202) was concentrated on the nitrogen atom than that on the oxygen atom (-0.180), indicating that the nitrogen attacking mode is more favorable for IM2-2, thus leading to the formation of N-heterocycle intermediate IM2-3. Furthermore, still by the Hirshfeld charges analysis of IM2-2'

		shown below, we found that the negative charge at the oxygen atom of the carbonyl group is -0.364, which means that oxygen has higher nucleophilicity than that of nitrogen of IM2-2', this would result in the generation of O-heterocycle intermediate IM2-3'. However, by comparing the Gibbs free energy of IM2-3 and that of IM2-3', we found that IM2-3 (-16.9 kcal/mol) is thermodynamically more stable than IM2-3' (-14.5 kcal/mol). Thus, N-attack mode is thermodynamically favorable in IM2-2 under reaction condition A. (Figure 2a).⁶⁰ Additionally, by the Hirshfeld charges analysis of IM3-2, we observed that more negative charge (-0.309) was concentrated on the oxygen atom of amide group than that on the nitrogen atom (-0.070), which indicates that the O-attack mode is more favorable for IM3-2. And, we respectively calculated the activation energy barrier for O-attack and N-attack in IM3-2. The O-attack pathway needs to overcome an activation energy barrier of 28.1 kcal/mol, while nitrogen atom accomplishes N-attack pathway with a higher barrier of 35.2 kcal/mol. The higher barrier of 35.2 kcal/mol makes it impossible for IM3-2 to undergo an N-attack mode under condition B. The comparison between Hirshfeld charge of the oxygen atom (-0.309) and that of nitrogen atom (-0.070) also illustrate that O-attack mode is more favorable for IM3-2. In summary, O-attack is kinetically favorable under condition B, leading to the generation of O-heterocycle intermediate IM3-3 under condition B. (Figure 2b).
--	--	--

Page 5 Figure 2		Page 5 Right column Paragraph 10	Based on the aforementioned mechanistic studies and the outcomes of the previous reports,^{36, 52, 54, 61, 62} we postulated a plausible mechanism for the formation of 2a and 3a, shown in Scheme 5. For the formation of product 2a (Scheme 5a), $\text{PhI}(\text{OMe})_2$ is first generated in situ from the reaction of PhIO with MeOH. Then complex A is formed by $\text{PhI}(\text{OMe})_2$ and TMSOTf, enabling the electrophilic reaction with substrate 1a, leading to the formation of iodonium ion B. Next, isomerization of the amide occurs via an intermolecular proton shift, and the subsequent nucleophilic attack of the nitrogen atom gives intermediate C.	Based on the aforementioned mechanistic studies and the outcomes of the previous reports,^{36, 52, 54, 61, 62} we postulated a plausible mechanism for the formation of 2a and 3a, shown in Scheme 5. For the formation of product 2a (Scheme 5a), $\text{PhI}(\text{OMe})_2$ is first generated in situ from the reaction of PhIO with MeOH. Then complex IM2-1 is formed by $\text{PhI}(\text{OMe})_2$ and TMSOTf, enabling the electrophilic reaction with isomerized substrate 1a', leading to the formation of iodonium ion IM2-2. Next, isomerization of the amide occurs via an intermolecular proton shift, and the subsequent nucleophilic attack of the nitrogen atom gives intermediate IM2-4.
Page 6 Scheme 3		Page 6 Left column Paragraph 10	Then the deprotonation by the methoxy anion forms intermediate D. The activated carbon atom bonded to iodine-center is nucleophilically	Then the deprotonation by the methoxy anion forms intermediate IM2-5. The activated carbon atom bonded to iodine-center is nucleophilically attacked by

	attacked by the phenyl ring, giving a phenonium ion E.⁶³⁻⁶⁷ Then, ring opening of the cyclopropane moiety in E occurs with simultaneous ring expansion to give carbocation F. Finally, the elimination reaction occurs in F to form isoquinolinone 2a. Similarly, with regard to the formation of product 3a (Scheme 5b), the modified Koser's reagent is first formed from the reaction of PhIO with ArSO₃H. It is worth noting that the coordination of the hydroxyl group to LiClO₄ results in the formation of a thermodynamically more stable complex G, which undergoes electrophilic addition with olefin to form intermediate H. Next, the nucleophilic attack of oxygen atom of the amide moiety onto the benzylic carbon center in H produces intermediate I. After that, deprotonation by the sulphonate ion enables the conversion of iminium I to imine J. Then the aromatic backbone of J undergoes 1,2-aryl migration^{36, 53, 68, 69} to form carbon cation K. Finally, with the removal of the acidic proton by the hydroxide anion released from the prior step, intermediate K is converted to iminoisocoumarin 3a is obtained.	the phenyl ring, giving a phenonium ion IM2-6.⁶³⁻⁶⁷ Then, ring opening of the cyclopropane moiety in IM2-6 occurs with simultaneous ring expansion to give carbocation IM2-7. Finally, the elimination reaction occurs in IM2-7 to form isoquinolinone 2a. Similarly, with regard to the formation of product 3a (Scheme 5b), the modified Koser's reagent is first formed from the reaction of PhIO with ArSO₃H. It is worth noting that the coordination of the hydroxyl group to LiClO₄ results in the formation of a thermodynamically more stable complex IM3-1, which undergoes electrophilic addition with olefin to form intermediate IM3-2. Next, the nucleophilic attack of oxygen atom of the amide moiety onto the benzylic carbon center in IM3-2 produces intermediate IM3-4. After that, deprotonation by the sulphonate ion enables the conversion of iminium IM3-4 to imine IM3-6. Then the aromatic backbone of IM3-6 undergoes 1,2-aryl migration^{36, 53, 68, 69} to form carbon cation IM3-7. Finally, with the removal of the acidic proton by the hydroxide anion released from the prior step, intermediate IM3-7 is converted to iminoisocoumarin 3a is obtained.
Page 6 Scheme 4	Scheme 4. Organocatalytic transformation of regiodivergent synthesis.	Scheme 4. Organocatalytic transformation of chemodivergent synthesis.
Page 6 Right column Paragraph 12	In summary, we have presented an exclusive regiodivergent cycloisomerization approach for constructing isoquinolinones and iminoisocoumarins skeletons starting from an identical o -alkenylbenzamides derivative.	In summary, we have presented an exclusive chemodivergent cycloisomerization approach for constructing isoquinolinones and iminoisocoumarins skeletons starting from an identical o -alkenylbenzamides derivative.

The end

REVIEWERS' COMMENTS:

Reviewer #2 (Remarks to the Author):

The revisions requested to this manuscript have been adequately performed and I am satisfied with the outcome.

There are a number of small errors in the manuscript, such as spaces after compound labels, etc, but these will be dealt with during typesetting and need not be addressed now.

I recommend this be accepted.

REVIEWERS' COMMENTS

Reviewer #2 (Remarks to the Author):

The revisions requested to this manuscript have been adequately performed and I am satisfied with the outcome.

There are a number of small errors in the manuscript, such as spaces after compound labels, etc, but these will be dealt with during typesetting and need not be addressed now.

I recommend this be accepted.

Response: We greatly appreciate the positive comments from this Reviewer. We have double checked the manuscript and had the corresponding errors corrected. Please see the Table starting on the next page for details.

Table. A Point-to-point Description of Changes

Location	Original MS	Revised MS
Page 1 Title	Concomitant 1,2-Aryl Migration/Elimination Reaction Mediated by Hypervalent Iodine Reagents: Chemoselective Cycloisomerization of O -Alkenylbenzamides	Chemoselective Cycloisomerization of O -alkenylbenzamides via Concomitant 1,2-Aryl migration/Elimination Mediated by Hypervalent Iodine Reagents
Page 1 Author name	Jiaxin He ^{1,3} , Feng-Huan Du ^{2,3} , Chi Zhang ^{2,*} & Yunfei Du ^{1,*}	Jiaxin He ¹ , Feng-Huan Du ² , Chi Zhang ^{2,*} & Yunfei Du ^{1,*}
Page 1 Author tagging statements	³ These authors contributed equally.	³ These authors contributed equally: Jiaxin He, Feng-Huan Du.
Page 1 Corresponding author's email	E-mail: duyunfeier@tju.edu.cn; zhangchi@nankai.edu.cn.	e-mail: zhangchi@nankai.edu.cn; duyunfeier@tju.edu.cn.
Page 1 Abstract	A chemodivergent cycloisomerization approach to construct isoquinolinone and iminoisocoumarin skeletons from o -alkenylbenzamide derivatives is established. The chemo-controllable strategy employed an exclusive 1,2-aryl migration/elimination cascade, enabled by different hypervalent iodine species generated in situ from the reaction of iodosobenzene (PhIO) with MeOH or 2,4,6-tris-isopropylbenzene sulfonic acid. DFT studies revealed that the nitrogen and oxygen atoms of the intermediates in the two reaction systems have different nucleophilicities and thus produce the selectivity of N or O -attack modes.	As an ambident nucleophile, controlling the reaction selectivities of nitrogen and oxygen atoms in amide moiety is a challenging issue in organic synthesis. Herein, we present a chemodivergent cycloisomerization approach to construct isoquinolinone and iminoisocoumarin skeletons from o -alkenylbenzamide derivatives. The chemo-controllable strategy employed an exclusive 1,2-aryl migration/elimination cascade, enabled by different hypervalent iodine species generated in situ from the reaction of iodosobenzene (PhIO) with MeOH or 2,4,6-tris-isopropylbenzene sulfonic acid. DFT studies revealed that the nitrogen and oxygen atoms of the intermediates in the two reaction systems have different nucleophilicities and thus produce the selectivity of N or O -attack modes.
Page 1	added heading	Introduction
Page 1	Herein, we present our results in controlling the reactivity of o -	Herein, we present our results in controlling the reactivity of o -

Right column Paragraph 2	alkenylbenzamides by hypervalent iodine species generated in situ .	alkenylbenzamides by hypervalent iodine species generated in situ .
Page 1 Left column Paragraph 1	In this regard, a new protocol dictating the chemodivergent cycloisomerization of o -alkenylbenzamides, by tuning the differentiation of the N vs O nucleophilic strength, to construct novel heterocyclic skeletons should be highly desirable.	In this regard, a new protocol dictating the chemodivergent cycloisomerization of o -alkenylbenzamides, by tuning the differentiation of the N vs O nucleophilic strength, to construct novel heterocyclic skeletons should be highly desirable.
Page 1 Paragraph 2	scheme 1a; scheme 1b; scheme 1c	Fig. 1a; Fig. 1b; Fig. 1c
Page 1 Right column	Scheme 1. Strategies for divergent cyclization of amide derivatives.	Fig. 1 Strategies for divergent cyclization of amide derivatives. a,b Previous works on cycloisomerization of amide derivatives. c This work, hypervalent iodine reagents mediated intramolecular cycloisomerization and 1,2-aryl migration.
Page 1 Right column Paragraph 3	When substrate 1a was treated with iodosobenzene (PhIO) in MeOH, combined with BF ₃ ·OEt ₂ (0.2 equiv) as a Lewis acid, we found the reaction displayed a completely distinct N -attack cyclization mode to furnish isoquinolinone 2a in 56% yield.	When substrate 1a was treated with iodosobenzene (PhIO) in MeOH, combined with BF ₃ ·OEt ₂ (0.2 equiv) as a Lewis acid, we found the reaction displayed a completely distinct N -attack cyclization mode to furnish isoquinolinone 2a in 56% yield.
Page 2 Left column Table 1	^[a] Reaction conditions: 1a (0.5 mmol), HIR (1.5 equiv), solvent (5.0 mL), rt. NR = no reaction. ^[b] Entries 3-12, additives (1.5 equiv) were used; entries 13-20, additives (0.2 equiv) were used. ^[c] Isolated yield. ^[d] BF ₃ ·OEt ₂ (1.5 equiv) was added. ^[e] TMSOTf (1.5 equiv) was added. ^[f] LiClO ₄ (1.5 equiv) was added. ^[g] Zn(ClO ₄) ₂ (1.5 equiv) was added. ^[h] LiClO ₄ (1.0 equiv) was added. ^[i] LiClO ₄ (1.8 equiv) was added. See SI For more details.	^[a] Reaction conditions: 1a (0.5 mmol), HIR (1.5 equiv), solvent (5.0 mL), rt. NR = no reaction. ^[b] Entries 3-12, additives (1.5 equiv) were used; entries 13-20, additives (0.2 equiv) were used. ^[c] Isolated yield. ^[d] BF ₃ ·OEt ₂ (1.5 equiv) was added. ^[e] TMSOTf (1.5 equiv) was added. ^[f] LiClO ₄ (1.5 equiv) was added. ^[g] Zn(ClO ₄) ₂ (1.5 equiv) was added. ^[h] LiClO ₄ (1.0 equiv) was added. ^[i] LiClO ₄ (1.8 equiv) was added. (For details see Supplementary Table S1, S2).
Page 2 Left column Paragraph 4	Encouraged by the above results, we then turned our attention toward exploring the chemodivergent pattern of the protocol for the synthesis of O -cyclized iminoisocoumarin products.	Encouraged by the above results, we then turned our attention toward exploring the chemodivergent pattern of the protocol for the synthesis of O -cyclized iminoisocoumarin products.

Page 2 Left column Table 1	   Entry^a HIR^b Solvent^c Additive^d T (°C)^e Yield^d of^e        2a (%) 3a (%)    1^aPhIO^bMeOH^cBF₃·OEt₂^drt^e56^d—^e 2^aPhIO^bMeOH^cEt₃N^drt^eNR^d—^e 3^aPhIO^bMeOH^cTFA^drt^e61^d—^e 4^aPhIO^bMeOH^cTfOH^drt^e63^d—^e 5^aPhIO^bMeOH^cTMSOTf^drt^e70^d—^e 6^aPhIO^bMeOH^cLiClO₄^drt^eNR^d—^e 7^aPhIO^bMeOH^c50% H₂SO₄^drt^e58^d—^e 8^aPhIO^bMeOH^cTMSOTf^dreflux^e81^d—^e 9^aHTIB^bDCE^c—^drt^e—^d44^e 10^aHTIB^bDCE^c—^d80^e—^d55^e 11^aPhIO^bDCE^cS1^d80^e—^d54^e 12^aPhIO^bDCE^cS2^d80^e—^d52^e 13^aPhIO^bDCE^cS3^d80^e—^d63^e 14^aPhIO^bDCE^cS4^d80^e—^d69^e 15^aPhIO^bDCE^cS4^d80^e—^d73^e 16^bPhIO^bDCE^cS4^d80^e—^d70^e 17^bPhIO^bDCE^cS4^d80^e—^d90^e 18^bPhIO^bDCE^cS4^d80^e—^d82^e 19^bPhIO^bDCE^cS4^d80^e—^d77^e 20^bPhIO^bDCE^cS4^d80^e—^d90^e  	Entry ^a	HIR ^b	Solvent ^c	Additive ^d	T (°C) ^e	Yield ^d of ^e							2a (%)	3a (%)	1 ^a	PhIO ^b	MeOH ^c	BF ₃ ·OEt ₂ ^d	rt ^e	56 ^d	— ^e	2 ^a	PhIO ^b	MeOH ^c	Et ₃ N ^d	rt ^e	NR ^d	— ^e	3 ^a	PhIO ^b	MeOH ^c	TFA ^d	rt ^e	61 ^d	— ^e	4 ^a	PhIO ^b	MeOH ^c	TfOH ^d	rt ^e	63 ^d	— ^e	5 ^a	PhIO ^b	MeOH ^c	TMSOTf ^d	rt ^e	70 ^d	— ^e	6 ^a	PhIO ^b	MeOH ^c	LiClO ₄ ^d	rt ^e	NR ^d	— ^e	7 ^a	PhIO ^b	MeOH ^c	50% H ₂ SO ₄ ^d	rt ^e	58 ^d	— ^e	8 ^a	PhIO^b	MeOH^c	TMSOTf^d	reflux^e	81^d	— ^e	9 ^a	HTIB ^b	DCE ^c	— ^d	rt ^e	— ^d	44 ^e	10 ^a	HTIB ^b	DCE ^c	— ^d	80 ^e	— ^d	55 ^e	11 ^a	PhIO ^b	DCE ^c	S1 ^d	80 ^e	— ^d	54 ^e	12 ^a	PhIO ^b	DCE ^c	S2 ^d	80 ^e	— ^d	52 ^e	13 ^a	PhIO ^b	DCE ^c	S3 ^d	80 ^e	— ^d	63 ^e	14 ^a	PhIO ^b	DCE ^c	S4 ^d	80 ^e	— ^d	69 ^e	15 ^a	PhIO ^b	DCE ^c	S4 ^d	80 ^e	— ^d	73 ^e	16 ^b	PhIO ^b	DCE ^c	S4 ^d	80 ^e	— ^d	70 ^e	17 ^b	PhIO^b	DCE^c	S4^d	80^e	— ^d	90^e	18 ^b	PhIO ^b	DCE ^c	S4 ^d	80 ^e	— ^d	82 ^e	19 ^b	PhIO ^b	DCE ^c	S4 ^d	80 ^e	— ^d	77 ^e	20 ^b	PhIO ^b	DCE ^c	S4 ^d	80 ^e	— ^d	90 ^e	   Entry^a HIR^b Solvent^c Additive^d T (°C)^e Yield^d of^e        2a (%) 3a (%)    1^aPhIO^bMeOH^cBF₃·OEt₂^drt^e56^d—^e 2^aPhIO^bMeOH^cEt₃N^drt^eNR^d—^e 3^aPhIO^bMeOH^cTFA^drt^e61^d—^e 4^aPhIO^bMeOH^cTfOH^drt^e63^d—^e 5^aPhIO^bMeOH^cTMSOTf^drt^e70^d—^e 6^aPhIO^bMeOH^cLiClO₄^drt^eNR^d—^e 7^aPhIO^bMeOH^c50% H₂SO₄^drt^e58^d—^e 8^aPhIO^bMeOH^cTMSOTf^dreflux^e81^d—^e 9^aHTIB^bDCE^c—^drt^e—^d44^e 10^aHTIB^bDCE^c—^d80^e—^d55^e 11^aPhIO^bDCE^cS1^d80^e—^d54^e 12^aPhIO^bDCE^cS2^d80^e—^d52^e 13^aPhIO^bDCE^cS3^d80^e—^d63^e 14^aPhIO^bDCE^cS4^d80^e—^d69^e 15^bPhIO^bDCE^cS4^d80^e—^d73^e 16^bPhIO^bDCE^cS4^d80^e—^d70^e 17^bPhIO^bDCE^cS4^d80^e—^d90^e 18^bPhIO^bDCE^cS4^d80^e—^d82^e 19^bPhIO^bDCE^cS4^d80^e—^d77^e 20^bPhIO^bDCE^cS4^d80^e—^d90^e  	Entry ^a	HIR ^b	Solvent ^c	Additive ^d	T (°C) ^e	Yield ^d of ^e							2a (%)	3a (%)	1 ^a	PhIO ^b	MeOH ^c	BF ₃ ·OEt ₂ ^d	rt ^e	56 ^d	— ^e	2 ^a	PhIO ^b	MeOH ^c	Et ₃ N ^d	rt ^e	NR ^d	— ^e	3 ^a	PhIO ^b	MeOH ^c	TFA ^d	rt ^e	61 ^d	— ^e	4 ^a	PhIO ^b	MeOH ^c	TfOH ^d	rt ^e	63 ^d	— ^e	5 ^a	PhIO ^b	MeOH ^c	TMSOTf ^d	rt ^e	70 ^d	— ^e	6 ^a	PhIO ^b	MeOH ^c	LiClO ₄ ^d	rt ^e	NR ^d	— ^e	7 ^a	PhIO ^b	MeOH ^c	50% H ₂ SO ₄ ^d	rt ^e	58 ^d	— ^e	8 ^a	PhIO ^b	MeOH ^c	TMSOTf ^d	reflux ^e	81 ^d	— ^e	9 ^a	HTIB ^b	DCE ^c	— ^d	rt ^e	— ^d	44 ^e	10 ^a	HTIB ^b	DCE ^c	— ^d	80 ^e	— ^d	55 ^e	11 ^a	PhIO ^b	DCE ^c	S1 ^d	80 ^e	— ^d	54 ^e	12 ^a	PhIO ^b	DCE ^c	S2 ^d	80 ^e	— ^d	52 ^e	13 ^a	PhIO ^b	DCE ^c	S3 ^d	80 ^e	— ^d	63 ^e	14 ^a	PhIO ^b	DCE ^c	S4 ^d	80 ^e	— ^d	69 ^e	15 ^b	PhIO ^b	DCE ^c	S4 ^d	80 ^e	— ^d	73 ^e	16 ^b	PhIO ^b	DCE ^c	S4 ^d	80 ^e	— ^d	70 ^e	17 ^b	PhIO^b	DCE^c	S4^d	80^e	— ^d	90^e	18 ^b	PhIO ^b	DCE ^c	S4 ^d	80 ^e	— ^d	82 ^e	19 ^b	PhIO ^b	DCE ^c	S4 ^d	80 ^e	— ^d	77 ^e	20 ^b	PhIO ^b	DCE ^c	S4 ^d	80 ^e	— ^d	90 ^e
Entry ^a	HIR ^b	Solvent ^c	Additive ^d	T (°C) ^e	Yield ^d of ^e																																																																																																																																																																																																																																																																																																																	
					2a (%)	3a (%)																																																																																																																																																																																																																																																																																																																
1 ^a	PhIO ^b	MeOH ^c	BF ₃ ·OEt ₂ ^d	rt ^e	56 ^d	— ^e																																																																																																																																																																																																																																																																																																																
2 ^a	PhIO ^b	MeOH ^c	Et ₃ N ^d	rt ^e	NR ^d	— ^e																																																																																																																																																																																																																																																																																																																
3 ^a	PhIO ^b	MeOH ^c	TFA ^d	rt ^e	61 ^d	— ^e																																																																																																																																																																																																																																																																																																																
4 ^a	PhIO ^b	MeOH ^c	TfOH ^d	rt ^e	63 ^d	— ^e																																																																																																																																																																																																																																																																																																																
5 ^a	PhIO ^b	MeOH ^c	TMSOTf ^d	rt ^e	70 ^d	— ^e																																																																																																																																																																																																																																																																																																																
6 ^a	PhIO ^b	MeOH ^c	LiClO ₄ ^d	rt ^e	NR ^d	— ^e																																																																																																																																																																																																																																																																																																																
7 ^a	PhIO ^b	MeOH ^c	50% H ₂ SO ₄ ^d	rt ^e	58 ^d	— ^e																																																																																																																																																																																																																																																																																																																
8 ^a	PhIO^b	MeOH^c	TMSOTf^d	reflux^e	81^d	— ^e																																																																																																																																																																																																																																																																																																																
9 ^a	HTIB ^b	DCE ^c	— ^d	rt ^e	— ^d	44 ^e																																																																																																																																																																																																																																																																																																																
10 ^a	HTIB ^b	DCE ^c	— ^d	80 ^e	— ^d	55 ^e																																																																																																																																																																																																																																																																																																																
11 ^a	PhIO ^b	DCE ^c	S1 ^d	80 ^e	— ^d	54 ^e																																																																																																																																																																																																																																																																																																																
12 ^a	PhIO ^b	DCE ^c	S2 ^d	80 ^e	— ^d	52 ^e																																																																																																																																																																																																																																																																																																																
13 ^a	PhIO ^b	DCE ^c	S3 ^d	80 ^e	— ^d	63 ^e																																																																																																																																																																																																																																																																																																																
14 ^a	PhIO ^b	DCE ^c	S4 ^d	80 ^e	— ^d	69 ^e																																																																																																																																																																																																																																																																																																																
15 ^a	PhIO ^b	DCE ^c	S4 ^d	80 ^e	— ^d	73 ^e																																																																																																																																																																																																																																																																																																																
16 ^b	PhIO ^b	DCE ^c	S4 ^d	80 ^e	— ^d	70 ^e																																																																																																																																																																																																																																																																																																																
17 ^b	PhIO^b	DCE^c	S4^d	80^e	— ^d	90^e																																																																																																																																																																																																																																																																																																																
18 ^b	PhIO ^b	DCE ^c	S4 ^d	80 ^e	— ^d	82 ^e																																																																																																																																																																																																																																																																																																																
19 ^b	PhIO ^b	DCE ^c	S4 ^d	80 ^e	— ^d	77 ^e																																																																																																																																																																																																																																																																																																																
20 ^b	PhIO ^b	DCE ^c	S4 ^d	80 ^e	— ^d	90 ^e																																																																																																																																																																																																																																																																																																																
Entry ^a	HIR ^b	Solvent ^c	Additive ^d	T (°C) ^e	Yield ^d of ^e																																																																																																																																																																																																																																																																																																																	
					2a (%)	3a (%)																																																																																																																																																																																																																																																																																																																
1 ^a	PhIO ^b	MeOH ^c	BF ₃ ·OEt ₂ ^d	rt ^e	56 ^d	— ^e																																																																																																																																																																																																																																																																																																																
2 ^a	PhIO ^b	MeOH ^c	Et ₃ N ^d	rt ^e	NR ^d	— ^e																																																																																																																																																																																																																																																																																																																
3 ^a	PhIO ^b	MeOH ^c	TFA ^d	rt ^e	61 ^d	— ^e																																																																																																																																																																																																																																																																																																																
4 ^a	PhIO ^b	MeOH ^c	TfOH ^d	rt ^e	63 ^d	— ^e																																																																																																																																																																																																																																																																																																																
5 ^a	PhIO ^b	MeOH ^c	TMSOTf ^d	rt ^e	70 ^d	— ^e																																																																																																																																																																																																																																																																																																																
6 ^a	PhIO ^b	MeOH ^c	LiClO ₄ ^d	rt ^e	NR ^d	— ^e																																																																																																																																																																																																																																																																																																																
7 ^a	PhIO ^b	MeOH ^c	50% H ₂ SO ₄ ^d	rt ^e	58 ^d	— ^e																																																																																																																																																																																																																																																																																																																
8 ^a	PhIO ^b	MeOH ^c	TMSOTf ^d	reflux ^e	81 ^d	— ^e																																																																																																																																																																																																																																																																																																																
9 ^a	HTIB ^b	DCE ^c	— ^d	rt ^e	— ^d	44 ^e																																																																																																																																																																																																																																																																																																																
10 ^a	HTIB ^b	DCE ^c	— ^d	80 ^e	— ^d	55 ^e																																																																																																																																																																																																																																																																																																																
11 ^a	PhIO ^b	DCE ^c	S1 ^d	80 ^e	— ^d	54 ^e																																																																																																																																																																																																																																																																																																																
12 ^a	PhIO ^b	DCE ^c	S2 ^d	80 ^e	— ^d	52 ^e																																																																																																																																																																																																																																																																																																																
13 ^a	PhIO ^b	DCE ^c	S3 ^d	80 ^e	— ^d	63 ^e																																																																																																																																																																																																																																																																																																																
14 ^a	PhIO ^b	DCE ^c	S4 ^d	80 ^e	— ^d	69 ^e																																																																																																																																																																																																																																																																																																																
15 ^b	PhIO ^b	DCE ^c	S4 ^d	80 ^e	— ^d	73 ^e																																																																																																																																																																																																																																																																																																																
16 ^b	PhIO ^b	DCE ^c	S4 ^d	80 ^e	— ^d	70 ^e																																																																																																																																																																																																																																																																																																																
17 ^b	PhIO^b	DCE^c	S4^d	80^e	— ^d	90^e																																																																																																																																																																																																																																																																																																																
18 ^b	PhIO ^b	DCE ^c	S4 ^d	80 ^e	— ^d	82 ^e																																																																																																																																																																																																																																																																																																																
19 ^b	PhIO ^b	DCE ^c	S4 ^d	80 ^e	— ^d	77 ^e																																																																																																																																																																																																																																																																																																																
20 ^b	PhIO ^b	DCE ^c	S4 ^d	80 ^e	— ^d	90 ^e																																																																																																																																																																																																																																																																																																																
Page 2 Right column Paragraph 4	First, a detailed screening of solvents, hypervalent iodine reagents and temperature was carried out (see the SI for complete details).	First, a detailed screening of solvents, hypervalent iodine reagents and temperature was carried out (Supplementary Table S2).																																																																																																																																																																																																																																																																																																																				
Page 2 Right column Paragraph 4	Next, almost identical results were obtained when using active hypervalent iodine species formed in situ from iodosobenzene (PhIO) and 4-toluenesulfonic acid (Table 1, entry 11).^{51, 52}	Next, almost identical results were obtained when using active hypervalent iodine species formed in situ from iodosobenzene (PhIO) and 4-toluenesulfonic acid (Table 1, entry 11).^{51, 52}																																																																																																																																																																																																																																																																																																																				
Page 2 Right column	Table 2. Substrate scope study for synthesis of isoquinolinones 2.^[a,b]	Fig. 2 Substrate scope study for synthesis of isoquinolinones 2. ^[a] Reaction conditions: 1 (0.5 mmol), PhIO (1.5 equiv) and TMSOTf (0.2 equiv) in MeOH (5.0 mL) reflux for 0.5–2 h. Isolated yield.																																																																																																																																																																																																																																																																																																																				
Page 2 Right column Paragraph 5	With the optimized conditions in hand, we began to explore the general applicability of the divergent transformation, targeting the N-cyclization by using variously substituted N-phenyl-2-alkenylbenzamides with MeOH as a reaction partner^{34, 53–57} and solvent (Table 2) being first studied.	With the optimized conditions in hand, we began to explore the general applicability of the divergent transformation, targeting the N-cyclization by using variously substituted N-phenyl-2-alkenylbenzamides with MeOH as a reaction partner^{34, 53–57} and solvent being first studied (Fig. 2).																																																																																																																																																																																																																																																																																																																				
Page 3 Left column	Table 3. Substrate scope study for synthesis of iminoisocoumarins 3.^[a,b]	Fig. 3 Substrate scope study for synthesis of iminoisocoumarins 3. ^[a] Reaction conditions: 1 (0.5 mmol),																																																																																																																																																																																																																																																																																																																				

		PhIO (1.5 equiv), 2,4,6-tris-isopropylbenzene sulfonic acid (S4; 1.5 equiv) and LiClO ₄ (1.5 equiv) in DCE (5.0 mL) at 80 °C for 0.1-0.5 h. Isolated yield.
Page 3 Left column Paragraph 6	Next, we came to explore the chemodivergent synthesis of iminoisocoumarins 3 by subjecting substrates 1 to conditions B. As shown in Table 3, o -alkenylbenzamides 1b–v bearing different alkyl substituted R ¹ group and the substituted phenyl ring could smoothly furnish the corresponding iminoisocoumarins 3b–v with sole chemoselectivity in satisfactory to excellent yields (73–92%). Notably, in contrast to the inferior performance of substrate 1g and 1h in N -cyclization mode reaction, iminoisocoumarin product 3g and 3h could be obtained in high yield mediated by the modified Koser's reagent.	Next, we came to explore the chemodivergent synthesis of iminoisocoumarins 3 by subjecting substrates 1 to conditions B (Fig. 3). O -alkenylbenzamides 1b–v bearing different alkyl substituted R ¹ group and the substituted phenyl ring could smoothly furnish the corresponding iminoisocoumarins 3b–v with sole chemoselectivity in satisfactory to excellent yields (73–92%). Notably, in contrast to the inferior performance of substrate 1g and 1h in N -cyclization mode reaction, iminoisocoumarin product 3g and 3h could be obtained in high yield mediated by the modified Koser's reagent.
Page 3 Left column Paragraph 7	In addition, we carried out some control experiments to ascertain the hypervalent iodine species that is responsible for promoting the transformation (Scheme 2). First, we monitored the reaction process of substrate 1a with PhIO in MeOH in the absence of TMSOTf, and it was found that the reaction did not occur (Scheme 2a). Furthermore, the reaction carried out in the absence of PhIO also completely inhibited the transformation of substrate 1a into product 2a . The two results suggested that both hypervalent iodine reagent and Lewis acid are key reagents that enabled the transformation to occur. Next, when LiClO ₄ was solely applied to the transformation of 3a , no reaction occurred either. On the basis of this result as well as the outcome of	In addition, we carried out some control experiments to ascertain the hypervalent iodine species that is responsible for promoting the transformation (Fig. 4). First, we monitored the reaction process of substrate 1a with PhIO in MeOH in the absence of TMSOTf, and it was found that the reaction did not occur (Fig. 4a). Furthermore, the reaction carried out in the absence of PhIO also completely inhibited the transformation of substrate 1a into product 2a (Fig. 4b). The two results suggested that both hypervalent iodine reagent and Lewis acid are key reagents that enabled the transformation to occur. Next, when LiClO ₄ was solely applied to the transformation of 3a , no reaction occurred either (Fig. 4c). On the basis of this result as well as the outcome of the initial attempt of using HTIB (Table 1,

	the initial attempt of using HTIB (Table 1, entry 9), we tentatively infer that LiClO ₄ could on one hand promote the formation of Koser's reagent, while on the other hand, coordinate with hydroxy group in the hypervalent iodine specie formed in situ .	entry 9), we tentatively infer that LiClO ₄ could on one hand promote the formation of Koser's reagent, while on the other hand, coordinate with hydroxy group in the hypervalent iodine specie formed in situ .
Page 3 Right column	Scheme 2. Control experiments.	Fig. 4. Control experiments. a Control experiment to verify the necessity of hypervalent iodine reagent and Lewis acid. b Control experiment to verify the effect of LiClO ₄ .
Page 3 Right column Paragraph 8	To gain insight into the mechanism and chemoselectivity of above systems, we performed density functional theory (DFT) calculations on the reaction of substrate 1a under conditions A and conditions B (Figure 1). As previous work shows, PhI(OMe) ₂ is generated in situ by PhIO and MeOH. ^{54, 56, 57} It is known that TMSOTf can be present as TMS ⁺ + TfO ⁻ in organic solvents. ^{58, 59} The calculation shows that the complex IM2-1 , which is formed by PhI(OMe) ₂ and TMSOTf, is thermodynamically favored over reagent 1 by 14.0 kcal/mol (Figure 1a).	To gain insight into the mechanism and chemoselectivity of above systems, we performed density functional theory (DFT) calculations on the reaction of substrate 1a under conditions A and conditions B (Fig. 5). As previous work shows, PhI(OMe) ₂ is generated in situ by PhIO and MeOH. ^{54, 56, 57} It is known that TMSOTf can be present as TMS ⁺ + TfO ⁻ in organic solvents. ^{58, 59} The calculation shows that the complex IM2-1 , which is formed by PhI(OMe) ₂ and TMSOTf, is thermodynamically favored over reagent 1 by 14.0 kcal/mol (Fig. 5a).
Page 3 Right column Paragraph 9	For another reaction (Figure 1b), it starts with the coordination of the oxygen to the LiClO ₄ which results in the formation of a thermodynamically more stable complex IM3-1 by 10.8 kcal/mol.	For another reaction (Fig. 5b), it starts with the coordination of the oxygen to the LiClO ₄ which results in the formation of a thermodynamically more stable complex IM3-1 by 10.8 kcal/mol.
Page 4	Figure 1. DFT-computed potential energy profile for the reaction of 1a under the conditions A and conditions B. Ar = 2,4,6-triisopropyl. (Standard state: 25 °C, 1 mol/L). For details see the Supplementary Information.	Fig. 5 DFT-computed potential energy profile for the reaction of 1a under the conditions A and conditions B. Ar = 2,4,6-triisopropyl. (Standard state: 25 °C, 1 mol/L). (For details see Supplementary data 4)

Page 5 Left column Paragraph 10	Nitrogen or oxygen-attack mode is supported by Hirshfeld charges analysis of IM2-2 and IM3-2, which is capable of predicting electrophilicity and nucleophilicity (Figure 2). By the Hirshfeld charges analysis of IM2-2, we observed that more negative charge (-0.202) was concentrated on the nitrogen atom than that on the oxygen atom (-0.180), indicating that the nitrogen attacking mode is more favorable for IM2-2, thus leading to the formation of N-heterocycle intermediate IM2-3. Furthermore, still by the Hirshfeld charges analysis of IM2-2' shown below, we found that the negative charge at the oxygen atom of the carbonyl group is -0.364, which means that oxygen has higher nucleophilicity than that of nitrogen of IM2-2', this would result in the generation of O-heterocycle intermediate IM2-3'. Thus, N-attack mode is thermodynamically favorable in IM2-2 under reaction condition A. (Figure 2a).⁶⁰	Nitrogen or oxygen-attack mode is supported by Hirshfeld charges analysis of IM2-2 and IM3-2, which is capable of predicting electrophilicity and nucleophilicity (Fig. 6). By the Hirshfeld charges analysis of IM2-2, we observed that more negative charge (-0.202) was concentrated on the nitrogen atom than that on the oxygen atom (-0.180), indicating that the nitrogen attacking mode is more favorable for IM2-2, thus leading to the formation of N-heterocycle intermediate IM2-3. Furthermore, still by the Hirshfeld charges analysis of IM2-2' shown below, we found that the negative charge at the oxygen atom of the carbonyl group is -0.364, which means that oxygen has higher nucleophilicity than that of nitrogen of IM2-2', this would result in the generation of O-heterocycle intermediate IM2-3'. Thus, N-attack mode is thermodynamically favorable in IM2-2 under reaction condition A. (Fig. 6a).⁶⁰
Page 5 Left column	Figure 2. The Hirshfeld charges analysis of IM2-3 and IM3-2.	Fig. 6. The Hirshfeld charges analysis of IM2-3 and IM3-2. Relative free energies and electronic energies are given in kcal/mol.
Page 5 Right column Paragraph 10	Additionally, by the Hirshfeld charges analysis of IM3-2, we observed that more negative charge (-0.309) was concentrated on the oxygen atom of amide group than that on the nitrogen atom (-0.070), which indicates that the O-attack mode is more favorable for IM3-2. And, we respectively calculated the activation energy barrier for O-attack and N-attack in IM3-2. The O-attack pathway needs	Additionally, by the Hirshfeld charges analysis of IM3-2, we observed that more negative charge (-0.309) was concentrated on the oxygen atom of amide group than that on the nitrogen atom (-0.070), which indicates that the O-attack mode is more favorable for IM3-2. And, we respectively calculated the activation energy barrier for O-attack and N-attack in IM3-2. The O-attack pathway needs to overcome an

	to overcome an activation energy barrier of 28.1 kcal/mol, while nitrogen atom accomplishes N-attack pathway with a higher barrier of 35.2 kcal/mol. The higher barrier of 35.2 kcal/mol makes it impossible for IM3-2 to undergo an N-attack mode under condition B. The comparison between Hirshfeld charge of the oxygen atom (-0.309) and that of nitrogen atom (-0.070) also illustrate that O-attack mode is more favorable for IM3-2. In summary, O-attack is kinetically favorable under condition B, leading to the generation of O-heterocycle intermediate IM3-3 under condition B. (Figure 2b).	activation energy barrier of 28.1 kcal/mol, while nitrogen atom accomplishes N-attack pathway with a higher barrier of 35.2 kcal/mol. The higher barrier of 35.2 kcal/mol makes it impossible for IM3-2 to undergo an N-attack mode under condition B. The comparison between Hirshfeld charge of the oxygen atom (-0.309) and that of nitrogen atom (-0.070) also illustrate that O-attack mode is more favorable for IM3-2. In summary, O-attack is kinetically favorable under condition B, leading to the generation of O-heterocycle intermediate IM3-3 under condition B. (Fig. 6b).
Page 5 Right column Paragraph 11	Based on the aforementioned mechanistic studies and the outcomes of the previous reports,^{36, 52, 54, 61, 62} we postulated a plausible mechanism for the formation of 2a and 3a, shown in Scheme 5. For the formation of product 2a (Scheme 5a), PhI(OMe)₂ is first generated in situ from the reaction of PhIO with MeOH. Then complex IM2-1 is formed by PhI(OMe)₂ and TMSOTf, enabling the electrophilic reaction with isomerized substrate 1a', leading to the formation of iodonium ion IM2-2. Next, isomerization of the amide occurs via an intermolecular proton shift, and the subsequent nucleophilic attack of the nitrogen atom gives intermediate IM2-4.	Based on the aforementioned mechanistic studies and the outcomes of the previous reports,^{36, 52, 54, 61, 62} we postulated a plausible mechanism for the formation of 2a and 3a (Fig. 7). For the formation of product 2a (Fig. 7a), PhI(OMe)₂ is first generated in situ from the reaction of PhIO with MeOH. Then complex IM2-1 is formed by PhI(OMe)₂ and TMSOTf, enabling the electrophilic reaction with isomerized substrate 1a', leading to the formation of iodonium ion IM2-2. Next, isomerization of the amide occurs via an intermolecular proton shift, and the subsequent nucleophilic attack of the nitrogen atom gives intermediate IM2-4.
Page 5 Right column	Scheme 3. Possible reaction mechanism.	Fig. 7 Possible reaction mechanism. a Reaction formation mechanism of isoquinolinone 2a. b Reaction formation mechanism of iminoisocoumarin 3a.
Page 6 Left column Paragraph 11	Similarly, with regard to the formation of product 3a (Scheme 5b), the modified Koser's reagent is first	Similarly, with regard to the formation of product 3a (Fig. 7b), the modified

	formed from the reaction of PhIO with ArSO ₃ H.	Koser's reagent is first formed from the reaction of PhIO with ArSO ₃ H.
Page 6 Left column Paragraph 12	In view of the atomic economy of utilizing catalytic hypervalent iodine species, we further investigated the strategy of combining aryl iodine and exogenous oxidant to generate hypervalent iodine species in situ , with a purpose of demonstrating the economic application of this transformation. ⁷⁰⁻⁷⁴	In view of the atomic economy of utilizing catalytic hypervalent iodine species, we further investigated the strategy of combining aryl iodine and exogenous oxidant to generate hypervalent iodine species in situ , with a purpose of demonstrating the economic application of this transformation (Fig. 8). ⁷⁰⁻⁷⁴
Page 6 Left column	Scheme 4. Organocatalytic transformation of chemodivergent synthesis.	Fig. 8 Organocatalytic transformation of chemodivergent synthesis. Experiment to demonstrate the economic application of this transformation.
Page 6 Left column Paragraph 13	In summary, we have presented an exclusive chemodivergent cycloisomerization approach for constructing isoquinolinones and iminoisocoumarins skeletons starting from an identical o -alkenylbenzamides derivative. Notably, the divergent synthesis employed an exclusive 1,2-aryl migration/elimination strategy, which is realized by utilizing the different hypervalent iodine species generated in situ . In the reaction processes, different hypervalent iodine species was found to play a crucial role in the selectivity switch from N to O -cyclization, with the PhI(OMe) ₂ species inducing N -attack and the modified Koser's reagent favoring the O -attack in the cyclization step. DFT studies demonstrated that nitrogen and oxygen atoms of the intermediates in the two reaction systems have different nucleophilicity and thus produce the selectivity of N or O -attack mode.	In summary, we have presented an exclusive chemodivergent cycloisomerization approach for constructing isoquinolinones and iminoisocoumarins skeletons starting from an identical o -alkenylbenzamides derivative. Notably, the divergent synthesis employed an exclusive 1,2-aryl migration/elimination strategy, which is realized by utilizing the different hypervalent iodine species generated in situ . In the reaction processes, different hypervalent iodine species was found to play a crucial role in the selectivity switch from N to O -cyclization, with the PhI(OMe) ₂ species inducing N -attack and the modified Koser's reagent favoring the O -attack in the cyclization step. DFT studies demonstrated that nitrogen and oxygen atoms of the intermediates in the two reaction systems have different nucleophilicity and thus produce the selectivity of N or O -attack mode.
Page 6 Right column	The combined organic layer was dried over anhydrous Na ₂ SO ₄ and the	The combined organic layer was dried over anhydrous Na ₂ SO ₄ and the solvent

Methods	solvent was removed in vacuo. The residue was purified by flash column chromatography on silica gel to afford target product 2. The combined organic layer was dried over anhydrous Na₂SO₄ and the solvent was removed in vacuo. The residue was purified by flash column chromatography on silica gel to afford target product 3.	was removed in vacuo. The residue was purified by flash column chromatography on silica gel (petroleum ether/ethyl acetate 10:1) to afford target product 2. The combined organic layer was dried over anhydrous Na₂SO₄ and the solvent was removed in vacuo. The residue was purified by flash column chromatography on silica gel (petroleum ether/ethyl acetate 50:1) to afford target product 3.
Page 6 Right column Data Availability	The data that support the findings of this study are available within the article and the Supplementary Information. Details about materials and methods, experimental procedures, characterization data, mechanistic studies, NMR spectra are available in the Supplementary Information. The crystallographic data for compound 2p and 3t can be obtained free of charge from the Cambridge Crystallographic Data Centre (CCDC) under reference number 2201660 (2p) and 2202945 (3t).	All data generated during this study are included in this article and Supplementary Information. Experimental procedure, conditions optimization and product characterization are provided in the Supplementary Information. The NMR spectra of all compounds are available in Supplementary Data 1. The X-ray crystallographic coordinates for structures reported in this Article have been deposited at the Cambridge Crystallographic Data Centre (CCDC), under deposition number 2201660 (2p, Supplementary Data 2) and 2202945 (3t, Supplementary Data 3), respectively. These data can be obtained free of charge from the Cambridge Crystallographic Data Centre via www.ccdc.cam.ac.uk/data_request/cif. DFT calculations are available in Supplementary Data 4.
Page 8 References	54. Zhen, X., Wan, X., Zhang, W., Li, Q., Zhang-Negrerie, D. & Du, Y. Synthesis of Spirooxindoles from N-Arylamide Derivatives via Oxidative C(sp²)-C(sp³) Bond Formation Mediated by PhI(OMe)₂ Generated in situ. Org. Lett. 21, 890-894, (2019).	54. Zhen, X., Wan, X., Zhang, W., Li, Q., Zhang-Negrerie, D. & Du, Y. Synthesis of Spirooxindoles from N-Arylamide Derivatives via Oxidative C(sp²)-C(sp³) Bond Formation Mediated by PhI(OMe)₂ Generated in situ. Org. Lett. 21, 890-894, (2019).

Page 8 References	56. Zhang, J., Jalil, A., He, J., Yu, Z., Cheng, Y., Li, G., Du, Y. & Zhao, K. Lactonization with Concomitant 1,2-Aryl Migration and Alkoxylation Mediated by Dialkoxyphenyl Iodides Generated in situ . Chem. Commun. 57 , 7426-7429, (2021).	56. Zhang, J., Jalil, A., He, J., Yu, Z., Cheng, Y., Li, G., Du, Y. & Zhao, K. Lactonization with Concomitant 1,2-Aryl Migration and Alkoxylation Mediated by Dialkoxyphenyl Iodides Generated in situ . Chem. Commun. 57 , 7426-7429, (2021).
Page 8 References	71. Ochiai, M., Takeuchi, Y., Katayama, T., Sueda, T. & Miyamoto, K. Iodobenzene-Catalyzed α -Acetoxylation of Ketones. In Situ Generation of Hypervalent (Diacyloxyiodo)benzenes Using m -Chloroperbenzoic Acid. J. Am. Chem. Soc. 127 , 12244-12245, (2005).	71. Ochiai, M., Takeuchi, Y., Katayama, T., Sueda, T. & Miyamoto, K. Iodobenzene-Catalyzed α -Acetoxylation of Ketones. In Situ Generation of Hypervalent (Diacyloxyiodo)benzenes Using m -Chloroperbenzoic Acid. J. Am. Chem. Soc. 127 , 12244-12245, (2005).
Page 8 Right column Acknowledgements	Y.D. acknowledges the National Natural Science Foundation of China (No. 22071175). We also thank Dr. Jun Xu, Dr. Yan Gao and Prof. Xiangyang Zhang [AIC, SPST/TJU] for providing the analysis support.	YF.D. acknowledges the National Natural Science Foundation of China (No. 22071175). We also thank Dr. Jun Xu, Dr. Yan Gao and Prof. Xiangyang Zhang [AIC, SPST/TJU] for providing the analysis support.
Page 8 Right column Author Contributions	J.H. and Y.D. conceived and designed the experiments. J. H. performed all the experiments, prepared the manuscript and supporting information. F.-H.,D. performed all the DFT calculations work and collated the calculated data. C.Z. and Y.D. directed the research and revised the manuscript.	JX.H. , and YF.D. conceived and designed the experiments. JX. H. performed all the experiments, prepared the manuscript and supporting information. F-H.D. performed all the DFT calculations work and collated the calculated data. C.Z. , and YF.D. directed the research and revised the manuscript.

The end